# On the Role of the North Equatorial Counter Current during a Strong El Niño

**David J. Webb**

National Oceanography Centre, Southampton SO14 3ZH, U.K.

*Correspondence to:* D.J.Webb (djw@noc.ac.uk)

**Abstract.** An analysis of archived data from the Nemo 1/12th degree global ocean model shows the importance of the North Equatorial Counter Current in the development of the strong 1982-1983 and 1997-1998 El Niños. The model results indicate that in a normal year the core of warm water in the NECC is diluted by the surface Ekman transport, by geostrophic inflow and by tropical instability waves. During the development of the 1982-1983 and 1997-98 El Niños, these processes had reduced effect at the longitudes of warmest equatorial temperatures and to the west. During the autumn of 1982 and 1997 the speed of the NECC was also increased by a stronger than normal annual Rossby wave. The increased transport of warm water by the NECC due to these changes resulted in warm water reaching the far eastern Pacific and appears to have been a major factor in moving the centre of deep atmospheric convection eastwards across the Pacific.

## 1 Introduction

Studies of the tropical Pacific often focus on the Equatorial Waveguide and the propagation of equatorial Kelvin waves generated by westerly wind events (i.e. Levine and McPhaden, 2016; Chen et al., 2016; Hu and Fedorov, 2017). The study reported here starts in a similar manner, focusing on the Waveguide. It uses data from a long run of the Nemo 1/12th degree computer model of the global ocean and starts by calculating the average sea surface temperatures in the equatorial band as a function of longitude and time.

During the strong El Niño events of 1982-1983 and 1997-1998 the results show warm water propagating eastwards from the Warm Pool region of the West Pacific across to the South American coastline. A different type of event, the warm pool El Niños or oscillations (Kug et al., 2009), are

seen in other years but these are limited to the western and central Pacific.

The strong El Niño events propagate eastwards at a speed of about $0.6 \, \mathrm{m \, s^{-1}}$. The equatorial Pacific is highly stratified, with the warmest water concentrated in the top 200 m, so a speed of $0.6 \, \mathrm{m \, s^{-1}}$ is comparable with the speed of a number of equatorial Kelvin wave modes whose first zero occurs similarly at around 200 m. There is therefore some justification in connecting the propagation of the warm features with the propagation of equatorial Kelvin waves.

Except that this is highly unlikely.

Simple waves, like equatorial Kelvin waves, transport momentum and energy but they cannot easily transport quantities like temperature and salinity, qualities associated with individual particles in the medium. Such advection can only occur if the waves are highly non-linear so that particle velocities are comparable with the phase velocity. This occurs in breaking waves and, to a lesser extent, in tidal bores but, as far as the author is aware, no one has reported evidence for an equivalent major feature in the near-surface layers of the equatorial Pacific.

In order to clarify the situation, model archived data is used to calculate the flux of warm water across 180°E and 240°E as a function of time during the period 1980 to 1985. This period includes the strong 1982-1983 El Niño.

The study by Evans and Webster (2014) showed that a sea surface temperatures (SST) greater than 28°C is required for the onset of widespread deep convection over the tropical ocean. They also showed that at times temperatures of over 29.5°C may be required. For this reason the study concentrates on water temperatures that exceed 28°C.

The analysis shows that during the 1982-1983 El Niño, the main flux of warm water in the model did not occur within the equatorial waveguide. Instead it occurs further north, at

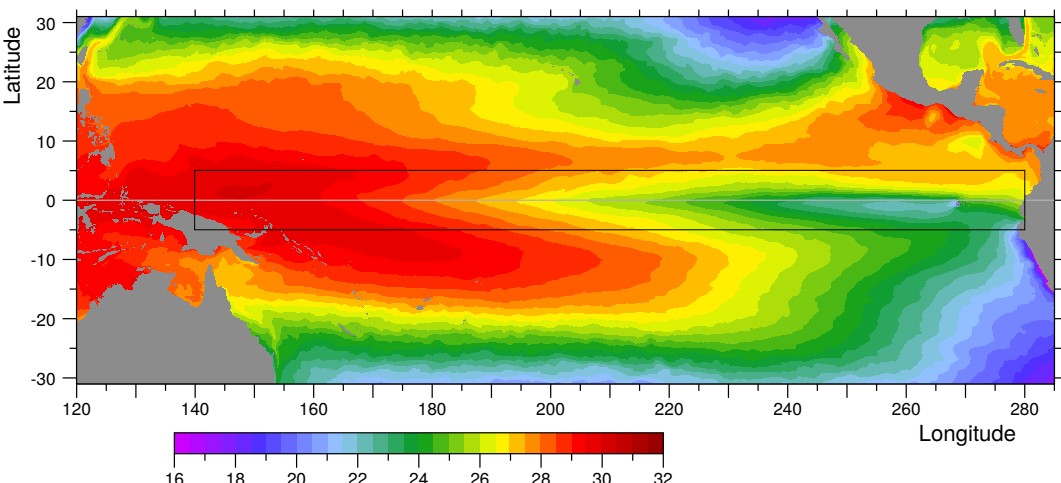

**Figure 1.** Average model sea surface temperature (°C) during 1981, showing the averaging region used for Figs. 5 and 6.

the latitude of the eastward flowing North Equatorial Counter Current (NECC).

Wyrtki (1973, 1974) was possibly the first to suggest that the NECC had the potential to transport significant amounts of heat eastwards in the Tropical Pacific. The NECC continues to be important in his later papers (i.e. Wyrtki, 1977, 1979) but in his theory of the El Niño (Wyrtki, 1975) he also introduced the idea of equatorial Kelvin waves triggering the El Niño. It is this aspect of his work that has been developed most by later authors.

To return to the NECC, the model results studied here indicate that in most years the Ekman transport, the geostrophic inflow and tropical instability waves carry warm water away from the core of the NECC and replace it with cooler water from the north and south. As a result the core temperature of the NECC is significantly reduced.

During periods when an El Niño is developing the trade winds retreat eastwards and they are replaced by a region of low or westerly winds. The model shows that one result of the low winds is that at the longitudes affected, the Ekman transport at the latitude of the NECC is reduced. The strength of the geostrophic inflow is also reduced as is the strength of the tropical instability waves. The latter is probably in part due to the reduction and change in direction of the surface current at the Equator.

As a consequence, while the El Niño develops, the NECC transports much warmer water than normal past the region of low winds. This transport of warm water occurs near the latitudes of the subtropical convergence in the atmosphere. Thus although the present study does not include an atmospheric model, it is likely that this is one, or possibly the main, factor moving the region of deep atmospheric convection and low winds further east. The process is then repeated, moving the convection region, the region of low winds and warmer than normal water, steadily eastwards across the ocean.

Further support for this argument is obtained by tracing particles during a strong El Niño and a non-El Niño year. The results from the model show that during the non-El Niño year, water particles are rapidly mixed out of the NECC but during the strong El Niño year they stay within the NECC and are transported further east.

The analysis also shows that the NECC is affected by annual Rossby waves which propagate westwards across the equatorial Pacific. These increase the speed of the NECC at all longitudes but, in particular, it is found that the wave at 6°N, arrives in the Western Pacific in mid-year, the time that the classic strong oceanographic El Niños usually start. In 1982 the amplitude of the wave in the Western Pacific was greater than normal and this may have started the strong 1982-1983 El Niño.

The structure of the paper is as follows. Section 2 describes the underlying numerical model and section 3 uses the results to plot the time series of average sea surface temperatures in the equatorial band during the period 1980 to 2000. Such a time series clearly shows the strong El Niño events of 1982-1983 and 1997-1998 as well as the weaker warm pool events in intermediate years.

Section 4 focuses on the eastward advection of warm water to determine when and where this occurs during the period 1980 to 1985. The NECC is found to be primarily responsible but there is a large year to year variability. Section 5 therefore then starts by examining the effects of Ekman transport, the geostrophic inflow and tropical instability waves on the the transport of warm water by NECC.

The section also investigates the varying strength of the NECC itself and this is developed further in in Section 6, which follows up Wyrtki's idea that El Niños are connected to the difference in sea level across the NECC.

Up to this point the analysis makes extensive use of Hovmöller diagrams, but to give a more geographical overview

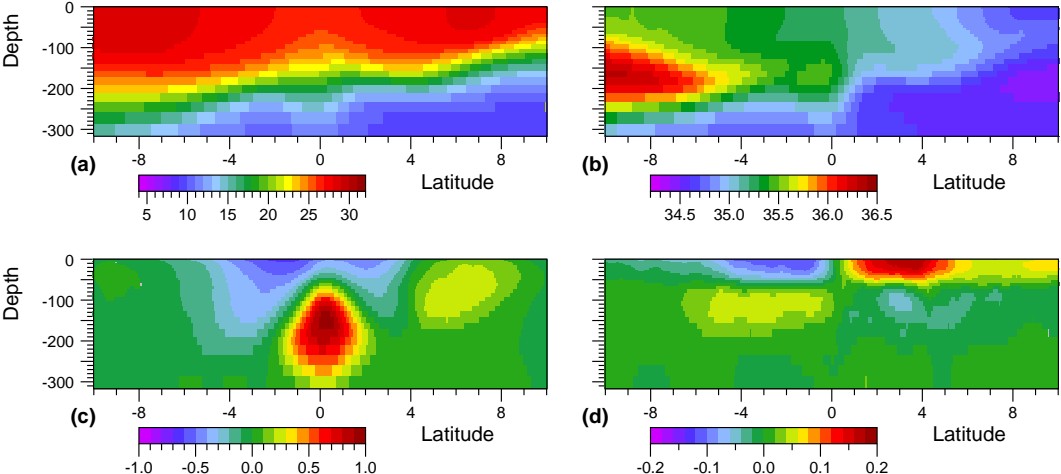

**Figure 2.** North-South Sections at 200°E (160°W) of the average values during 1981 of (a) temperature (°C), (b) salinity, and (c) east and (d) north components of velocity ($\mathrm{m\,s^{-1}}$).

of events, section 7 relates the results to plots of sea surface temperature, elevation and currents and the wind stress vectors, in the northern spring, summer, autumn and winter of 1982. Section 8 then investigates the mixing processes using particle tracks started in central Pacific the autumn of 1981 and 1982.

The final analysis section briefly reports on a similar analysis of the strong 1997-1998 El Niño. Although this El Niño starts differently, the role of the NECC, the annual Rossby wave and mixing processes is found to be similar to the 1982-1983 period. The paper closes with a review of the main results of the study.

## 2 The Nemo 1/12° global ocean model

The ocean model discussed here is one of the family of Nemo models (Madec, 2014), all with a similar code base but with different choices of horizontal and vertical grid resolution and of the many options for representing the underlying ocean physics. The present model uses a non-uniform grid based on a longitude grid spacing of 1/12° along the Equator. In the Southern Hemisphere and in the Indian and Pacific Oceans the latitude spacing is chosen so that each of the grid boxes has the same width and height. In the North Atlantic and Arctic a more complex scheme is used to prevent the convergence of the grid near the pole.

The model has 75 layers in the vertical. Their nominal thicknesses range from 1 m at the surface to 204 m in the lowest layer but as the ocean surface moves up and down each of the layers expands or contracts slightly to allow for this. The nominal thicknesses are based on an analytic formula which ensures a smooth transition between the strongly stratified surface layers, which need to be well resolved, and the weakly stratified deep ocean for which less resolution is

necessary. One consequence of this is that 35 layers are used to resolve just the top 300 m. In the equatorial Pacific this covers most of the major current systems.

This high resolution version of the Nemo ocean model was developed for coupling with a similar high-resolution version of the UK Meteorological Office atmospheric model, the aim being to create an improved coupled model for both weather prediction and climate change research. However before coupling the two models, a number of test runs were carried out using just the ocean model. The data analysed here comes from run 6, the last and longest of these tests.

The surface boundary conditions used for run 6 are those of Large and Yeager (2004) together with the Drakker DFS5.2 atmospheric fields described by Dussin et al. (2014). The Drakkar datasets, like the ECMWF reanalysis datasets on which they are based, start from 1958. This is also the start date of run 6.

In a previous analysis of run 6, Webb (2016) compared observed temperatures in the equatorial Pacific Niño regions with those from the model. The Niño regions are a series of standard ocean areas in the equatorial Pacific often used in El Niño studies (Trenberth, 1997).

This analysis showed that there was good agreement between observations and the model. It also showed that this was not due to the existence of a strong feedback loop - the actual sea surface temperature somehow controlling the model sea surface temperature (SST) via its effect of the atmospheric layers closest to the ocean surface.

On the basis of this analysis, and the additional confidence in the model code which comes from its successful use over many years for oceanic and climate research, it appears reasonable to make further use of the model archive data in the present study of processes affecting the El Niño.

Data from the model is available in the form of averages over each five-day period during the model run. The analysis reported here concentrates on 1981, as a typical non-El Niño year, and on the 1982-1983 and 1997-1998 periods, during which strong El Niños developed. The five day averages miss the diurnal variations in sea surface properties which may help to trigger convection. Observations from the west Pacific (King et al., 1991, Fig. 3) show temperature variations of one to two degrees during calm conditions but this resulted in only a few convective clouds. During windier, cloudier conditions the diurnal variation was usually less than a degree.

As background, Fig.1 shows the model average SST during 1981. In the west, the North Pacific Warm Pool shows a large region with average temperatures above 28°C. Similar temperatures are also found over a large region of the South Pacific. The figure also shows the region of the Equatorial Waveguide, extending from 5°S to 5°N, which is analysed in the next section of this paper. In the west this has average temperatures above 28°C, but in the east, where the Equatorial Undercurrent outcrops temperatures fall below 23°C.

Figure 2 shows the average values of temperature, salinity and velocity near the surface in a section at 200°E (160°W). The figure illustrates the strong stratification of the surface layers and the fact that the primary currents of the region, the Equatorial Current, the Equatorial Undercurrent and the North Equatorial Counter Current all lie close to the ocean surface.

The archive datasets also contain the ocean surface wind stress and precipitation fields used to force the model. The wind stress field (Fig. 3) shows that in the central Pacific during the second half of 1992, the normal trade winds, which extend to the equator, retreated eastwards. They were replaced by a region of low winds and, near the dateline, periods of westerly winds. On inspection the westerlies were often associated with cyclonic flows, probably due to convection, over warm water to the north and south of the Equator. The low wind stress region can also be seen in Figs. 22 to 25 from 1982 and Figs. 33 to 35 from 1997.

Figure 3 also shows that in the west Pacific, the winds are often light or westerly and that the strongest westerly winds often occur on either side of New Year. Plotted on a geographical grid, the wind stress vectors show that this is often due to winds converging on the South Pacific Convergence Zone. An example can be seen in Fig. 35.

The eastern limit of the low wind region along the equator appears to be associated with the eastern limit of the main deep convection region in the atmosphere and the associated region of high precipitation rates. However the forcing data shows that often there is limited precipitation near the Equator even when the wind stress is low.

However under these conditions there are usually bands of strong wind stress convergence to the north and south of the Equator close to areas with high rates of precipitation. Fig. 4 allows for this wider precipitation region by plotting the average precipitation rate between 12°S and 12°N, as a function

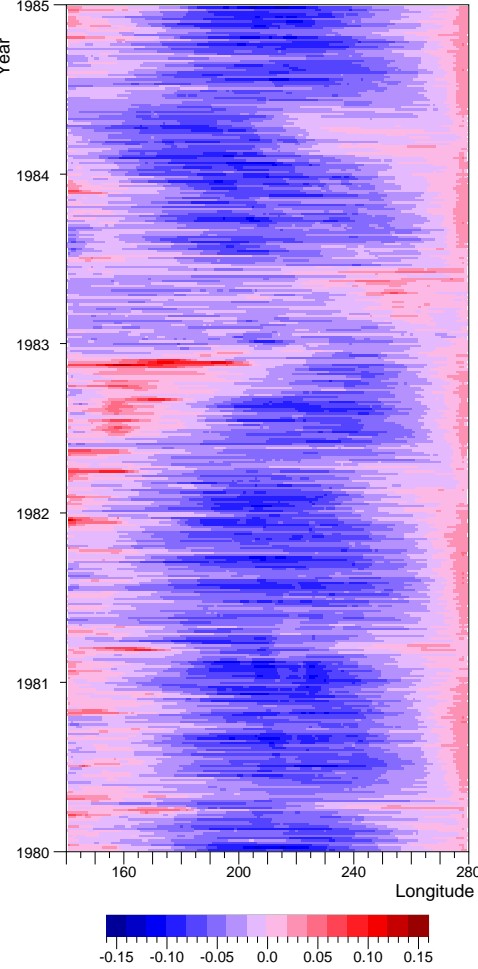

**Figure 3.** Eastward component of wind stress (Pa) on the Equator between 140°E and 280°E (80°W).

of longitude and time. The figure shows that during the second half of 1982 precipitation moved from the western to the central Pacific. The central Pacific values then declined early in 1983 after which there was a short period of increased precipitation near the eastern boundary. The main precipitation then moved back to the western Pacific.

In the South Pacific, precipitation may be enhanced by the South Pacific Warm Pool, especially late in the year after the Sun has crossed the Equator. When precipitation increases in the Eastern Pacific it often occurs along the line of the Inter Tropical Convergence Zone (ITCZ).

## 3   Time series of mean temperatures in the equatorial band

In the standard picture of an El Niño, warm water from the Western Pacific moves to the central equatorial Pacific and, in the more extreme cases all the way to the South Ameri-

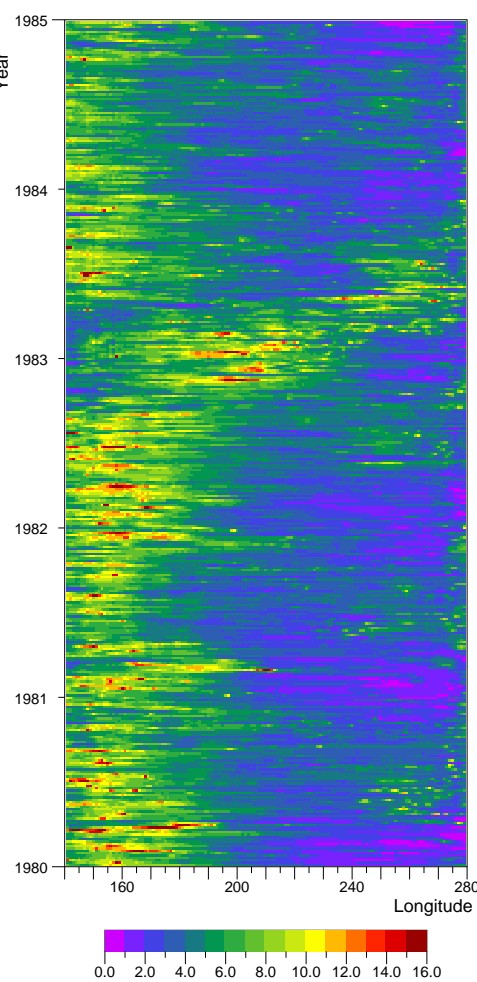

**Figure 4.** Precipitation $(\mathrm{kg\,m^{-2}d^{-1}})$ averaged between 12°S and 12°N.

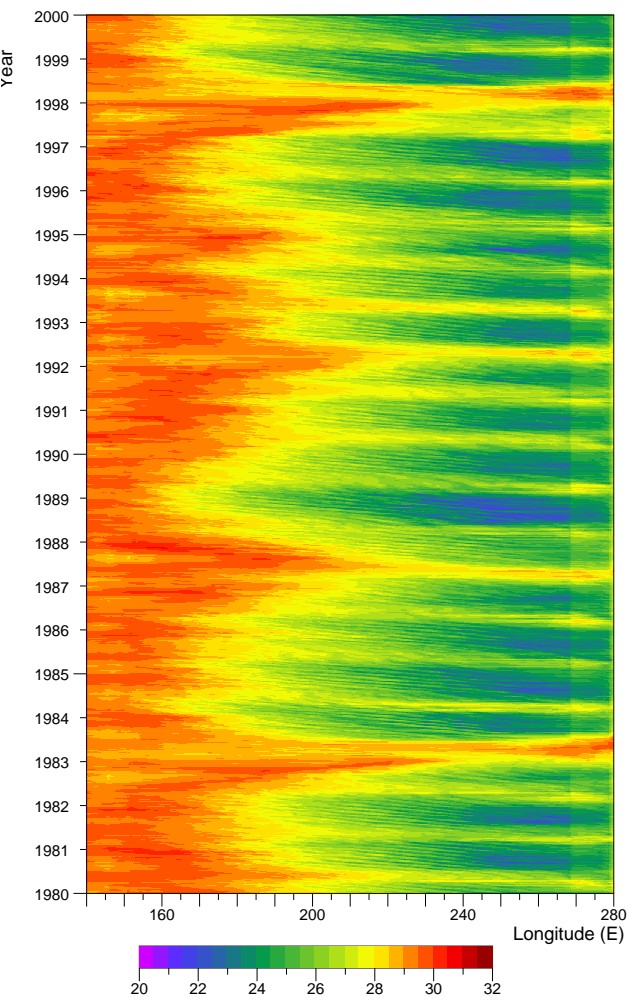

**Figure 5.** Average model sea surface temperature (°C) in the equatorial Pacific between 5°S and 5 °N during the period 1980.0 to 2000.0.

can coastline. This warming of the central and eastern Pacific moves the location of the dominant atmospheric convection region eastwards and, because of the amount of heat energy involved, results in large scale changes in the atmospheric circulation.

Figure 5 plots the sea surface temperature, averaged between 5°S and 5°N, as a function of longitude and for the period 1980 to 2000. During the whole of this period temperatures in the western equatorial Pacific hover around 30°C but in the eastern Pacific the mean temperature can vary between 22°C and 30°C.

The longest periods of warm temperatures in the eastern Pacific occur during the strong El Niños of 1982-83 and 1997-98 when the area is affected by features that have propagated in from the western Pacific. These El Niños are also abrupt, the warm temperature fronts extending rapidly all across the equatorial ocean, and then equally rapidly retreating to the far west.

Weaker warm pool events (Kug et al., 2009), are observed in 1987 and in 1991-1992 but these mainly affect surface temperatures in the central and western Pacific. They are also much more incremental, the temperature front only moving gradually east during the periods 1985 to 1987 and 1989 to 1992. Following 1992, the front only retreats to near its 1991 position.

The period between 1980 and 1985 is expanded in Fig. 6, together with a similar figure for the period including the 1997-1998 El Niño, emphasising the similarities between the two strong El Niños. The third figure comes from a similar period from a fully coupled run, where the same ocean model was coupled to a high resolution atmospheric model. In this case the ocean temperatures are slightly too low, but it shows that the detailed structures seen in the figures are not the result of forcing by, possibly anomalous, surface bound-

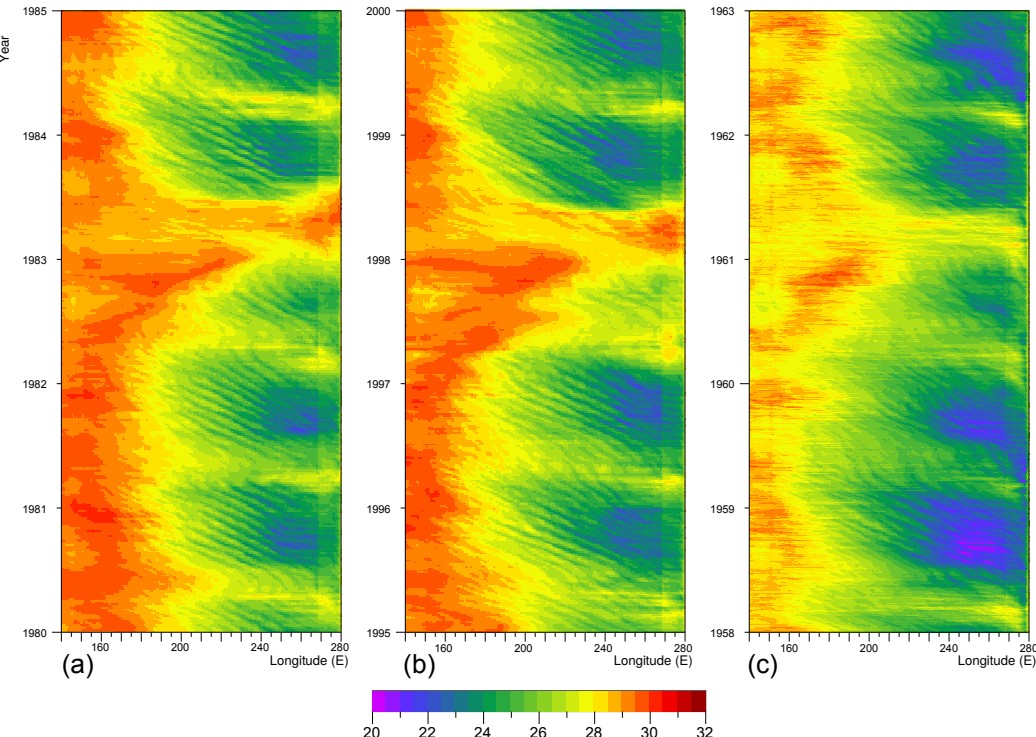

**Figure 6.** Expanded view of model sea surface temperature, averaged between 5°S and 5 °N, for the equatorial Pacific between 120°E and 280°E (80°W), and (a) between 1980 and 1985, (b) between 1995 and 2000 and (c) when coupled to a high resolution atmospheric model.

ary conditions but arise as natural variabilities of the coupled system.

As well as the main El Niño event, each of the figures show a series of fine scale wave-like features, with an east-west wavelength of five to ten degrees and a period of about a month. The features grow in amplitude during the autumn of each year and die out in the spring when their westward phase velocity tends to be reduced or even reversed. Their wavelength and period is consistent with them being due to the passage of tropical instability waves (TIWs). As each TIW passes it advects warm water into the equatorial band but after it has passed upwelling at the Equator will return the sea surface temperature to its earlier value.

The second feature of note is the annual signal which primarily affects the Eastern and Central Pacific. Near the South American coastline, temperatures are at a maximum early each year, a comparison with the wind field (i.e. Fig 3) indicating that this occurs after periods when the eastward component of the wind stress has dropped to near zero. Thus they are probably partly due to a reduction in equatorial upwelling. However this is also the time when the area of warm water off Central America has moved furthest south. An example is shown in Fig. 25.

Further west the period of low winds occurs later and this may explain why the temperature maximum in the annual signal occurs later towards the Central Pacific. Alternatively

it may be due to an annual wave triggered by the changes in the east.

Each of these features warrants further study, but in the rest of this paper the focus is on the the strong El Niño events when temperatures of 29°C and more are found all across the ocean. In a normal year, as illustrated in Figs. 5 and 6, the average temperatures in the Equatorial Waveguide, between 220°E and 240°E, lie between 24°C and 27.5°C. As shown in Fig. 1, warmer temperatures are found at 8°N, but on average these lie below 28°C and there is not enough heat available locally to explain the warming of the whole of the Eastern Pacific.

An increase in the local surface heat flux might produce the observed increases in model SST. However Webb (2016, see Figs. 11 and 13) investigated these fluxes in two of the key Niño regions and found that in each case the net heating was less during the development of the 1982-1983 El Niño than in the two previous years. Significant errors in the models vertical mixing scheme could also be responsible but this is unlikely. Instead it is much more probable that the increases in both the model SST and observations are due to advection of heat by the ocean.

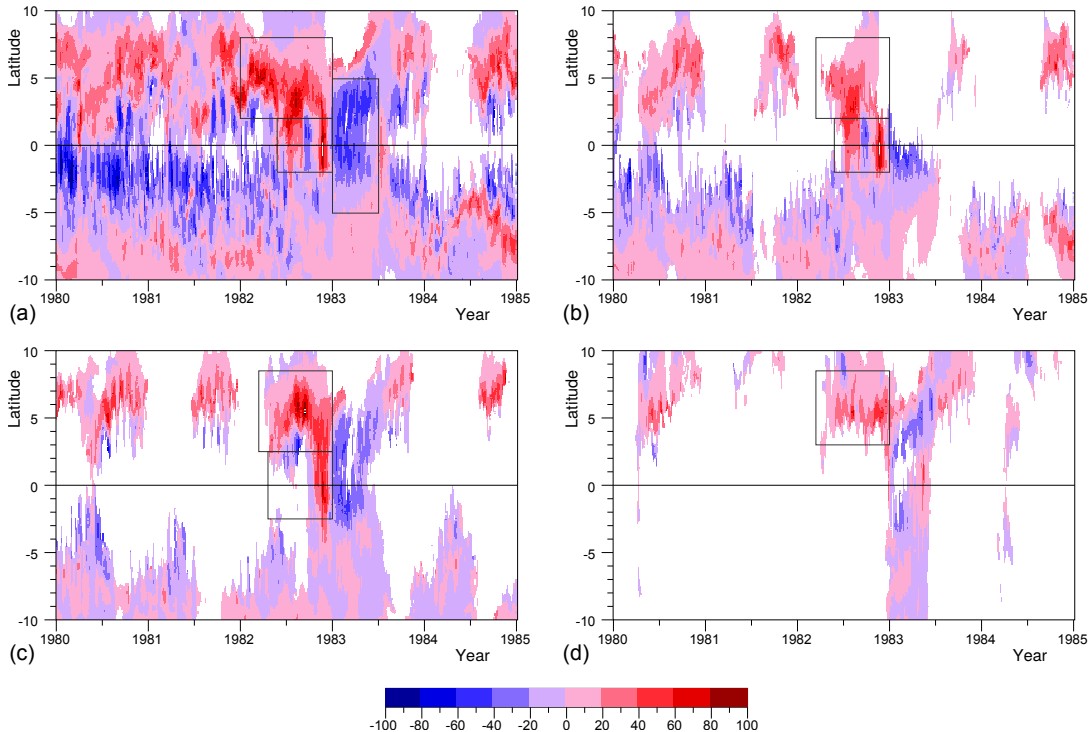

**Figure 7.** Vertically integrated flux of water ($\mathrm{m^2 s^{-1}}$) with temperature (a) greater than 28°C crossing longitude 180°E, (b) greater than 29°C crossing 180°E, (c) greater than 28°C crossing 210°E, (d) greater than 28°C crossing 240°E. The figure is blank where the flux is zero. Rectangles enclose the regions of table 1.

## 4   Zonal advection of warm water near the Equator

In an attempt to clarify how heat was advected in the equatorial Pacific, the integrated flux of water across a series of longitudes was calculated, with the constraint that the temperature had to be greater than an given minimum value. Figure 7 shows the results plotted as a time series at longitudes 180°E, 210°E and 240°E, for a minimum temperature of 28°C and, at longitude 180°E, for a minimum temperature of 29°C. The figure also includes a series of boxes, covering time periods and latitude ranges of interest, the total flux in each period being summarised in Table 1.

The figure shows that most of the eastward flow of warm water occurs at the latitudes of the North Equatorial Counter Current. The fluxes are largest in the autumn of each year, the temperatures in the spring occasionally being below the minimum temperature. The largest transports occur during 1982, in the period when the 1982-83 El Niño is developing. Thus Figs. 5a and 5b shows stronger than normal flows of 28+°C water at 180°E over most of 1982, and of 29+°C in late summer. At 210°E and 240°E the flow starts later in 1982 and continues until the year end.

In the equatorial band there are long periods when the water at all depths is too cool to contribute to the flux calculations. Longitude 180°E is an exception for a minimum temperature of 28°C, but the flow is predominantly westwards, as is expected for the latitudes of the Equatorial Current. At 180°E and 210°E, large eastward fluxes are observed during 1982, a small event occurring in late summer and a major event just before the end of the year. The major event is also seen at 210°E but is missing at 240°E. However it does show up when the minimum temperature is reduced to 26°C.

South of the Equator an eastward transport of warm water is also seen at the latitudes of the South Equatorial Counter Current. This is unexpected as, east of 180°E, the westward flowing South Equatorial Current is usually thought to be contiguous with the westward flowing Equatorial Current. However the flow is weak and reversing, and does not appear to be connected with the 1982-83 El Niño, so it is not considered further here.

Table 1 shows that at 210°E, the NECC transports a total of $310 \times 10^{12} \mathrm{m^3}$ of water warmer than 28°C between the spring and end of 1982. To give an idea of the potential impact of this volume of water, the table also shows the corresponding span of longitude that it would cover if it was contained in a surface layer 100 m thick extending from 5°S to 5°N.

The table also includes the contribution of the equatorial band for the same longitude and over roughly the same time period. Although the El Niño is often described as result-

| Region | Longitude | Minimum Temperature | Southern Boundary | Northern Boundary | Start Year | End Year | Net Flux $10^{12}\mathrm{m}^3$ | Longitude Span |
|--------|-----------|---------------------|-------------------|-------------------|------------|----------|-------------------------------|----------------|
| A | 180°E | 28°C | 2.0°N | 8.0°N | 1982.2 | 1983.0 | 484 | 39.2 |
| B |       |      | 2.0°S | 2.0°N | 1982.4 | 1983.0 | 124 | 10.0 |
| C |       |      | 5.0°S | 5.0°N | 1983.0 | 1983.5 | -417 | -33.8 |
| D | 180°E | 29°C | 2.0°N | 8.0°N | 1982.2 | 1983.0 | 217 | 17.6 |
| E |       |      | 2.0°S | 2.0°N | 1982.3 | 1983.0 | 119 | 9.6 |
| F | 210°E | 28°C | 2.5°N | 8.5°N | 1982.2 | 1983.0 | 310 | 25.1 |
| G |       |      | 2.5°S | 2.5°N | 1982.3 | 1983.0 | 79 | 6.4 |
| H | 240°E | 28°C | 2.0°N | 8.0°N | 1982.2 | 1983.0 | 168 | 13.6 |

**Table 1.** Volume transports of water for the longitudes, temperature classes, latitude bands and time periods denoted in Fig 7. The net flux is given both in units of $10^{12}\mathrm{m}^3$ and in terms of the number of degrees longitude that would be covered by the same volume if it was in a layer 100 m thick which extended from 5°S to 5°N.

ing from increased eastward heat advection in the Equatorial Waveguide, the contribution of the NECC is seen to be roughly four times larger than the contribution from currents close to the Equator. This is also true for water warmer than 28°C at 180°E.

In each of the three longitudes shown, the flux of water warmer than 28°C is not enough for a layer 100(m) thick to extend from 5°S to 5°N and all the way to South America (at 270°E), but it is sufficient to have a significant impact.

In summary, prior to the peak of the 1982-1983 El Niño, sufficient warm water was advected by the North Equatorial Counter Current and the Reversed Equatorial Current to produce significant warming of the eastern equatorial Pacific. Although it was not possible to to provide a full heat budget[1] there appears to be no reason to look for any other mechanism advecting warm nutrient poor waters into the eastern equatorial Pacific prior to an El Niño event.

Whereas most discussions of the El Niño focus on the role of the Equatorial Waveguide, these results show that, in the model, strong eastern Pacific El Niño events, like the 1982-83 El Niño, occur primarily as a result of heat transported by the North Equatorial Counter Current. Given the good agreement between the model and observations, discussed earlier, this is also likely to be true for the real ocean.

## 5   Heat transport and the NECC

These results raise the question of why the NECC transports so much heat in an El Niño year? Alternatively because, as Wyrtki (1975) pointed out, the NECC has its source in the West Pacific Warm Pool, the real question is "Why does the NECC transport so little heat in a non-El Niño year?".

Here it is argued that the transport is reduced by the combined effect of the Ekman transport, the geostrophic inflow, both parts of the tropical cell, and tropical instability waves.

[1]This is because of problems in reproducing the model mixing using only the data available in the 5-day average data sets.

All of these processes have the ability to remove warm water from the core of the NECC and to replace it by cooler water from the north or south.

Unfortunately our theoretical understanding of these three components and the NECC itself is poor. As a result the argument made here has to be based on a mixture of theory and analysis of the model results.

### 5.1   The Tropical Cell

This is probably the best understood part of the problem. As discussed by Stommel (1960), an easterly wind acting along the Equator produces a rise in sea level on the western boundary of the ocean. This results in a pressure gradient along the Equator which, when a steady state has been reached, exactly balances the surface wind stress.

If $\tau$ is the wind stress, $p$ the pressure in the ocean, $x$ the distance east and $z$ the depth, then this balance is given by,

$$\tau = \int dz\, \partial p/\partial x. \tag{1}$$

This pressure gradient also affects the upper layers of the ocean north and south of the equator, where it results in a geostrophic flow $v_g$ towards the Equator,

$$\rho f \int dz\, v_g = \int dz\, \partial p/\partial x,$$
$$= \tau, \tag{2}$$

where $\rho$ is density and $f$ the Coriolis term.

As the east-west pressure gradient changes only slowly with latitude, close to the Equator it can be considered as a constant, which means that the integral of $v_g$ tends to plus or minus infinity as $1/f$ as the Equator is approached.

Away from the Equator the surface wind stress generates an Ekman transport, $v_e$, such that,

$$-\rho f \int dz\, v_e = \tau. \tag{3}$$

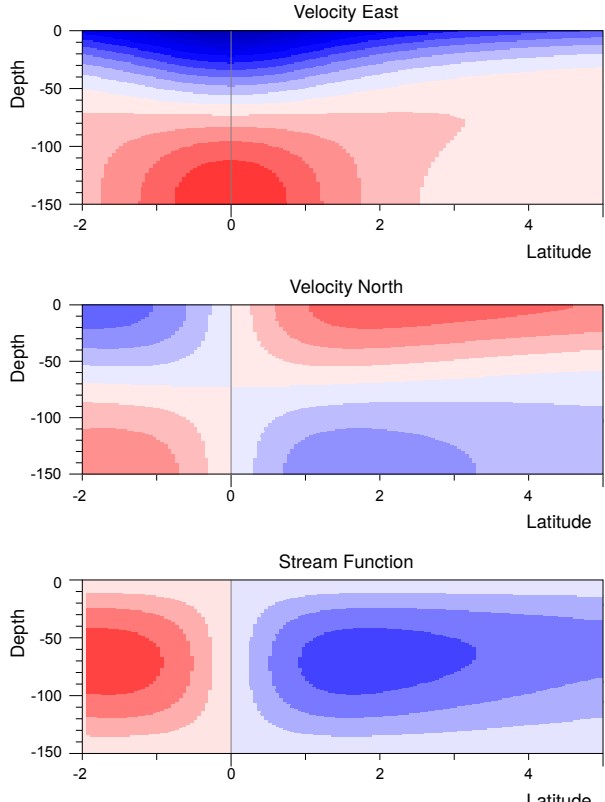

**Figure 8.** Solution of Stommel's model of the tropical cell, for a surface layer 150 m deep, of constant density and with a vertical kinematic viscosity of 100 $cm^2s^{-1}$ (Webb, 2018). Velocity contours at intervals of 5 $cm\,s^{-1}$. Stream function contours at intervals of 2 $m^2s^{-1}$. Positive values in red.

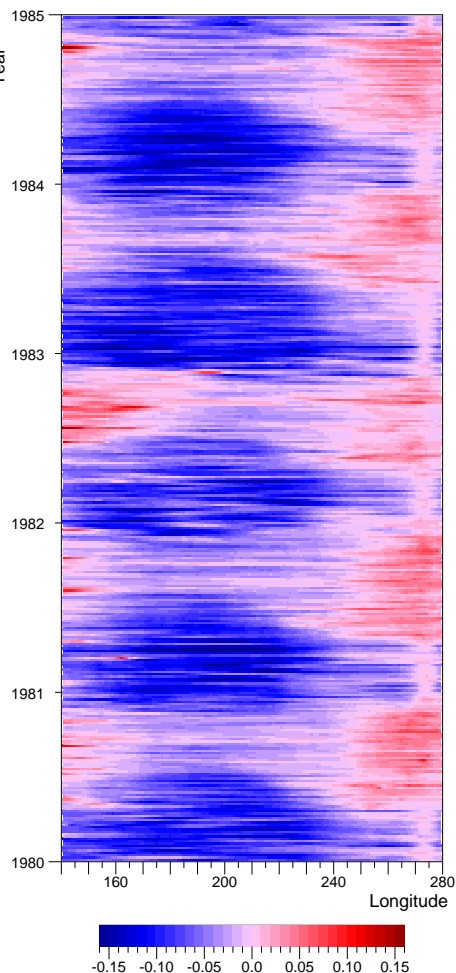

**Figure 9.** Eastward component of wind stress (Pa) at 6°N in the equatorial Pacific between 140°E and 280°E (80°W), between 1980 and 1985.

This integral also tends to plus or minus infinity as the equator is approached. However the singularity in $v_e$ exactly balances that in $v_g$, so as shown in Fig. 8, overall the solution is well behaved. In fact it is so well behaved that as well as the Ekman transport away from the Equator and the geostrophic inflow towards the Equator, the solution also includes the Equatorial Current and the Equatorial Undercurrent.

Unfortunately the theory has a major flaw. Stommel (1960) treated the ocean surface layer as one of constant density and with a constant vertical viscosity. He then found that if the vertical viscosity was reduced enough to generate a realistic undercurrent speed then the width of the undercurrent was only a fraction of a degree, whereas in reality it is a few degrees wide. If the viscosity is increased by a factor of 10, as it has been for Fig 8, then a reasonable width can be obtained but the maximum undercurrent velocity is only 25 $cm\,s^{-1}$ instead of a more realistic 150 $cm\,s^{-1}$.

The solution to the problem was eventually found by McCreary (1981) who showed that it was necessary to introduce stratification. When this was included, upwelling near the equator was limited by the rate at which heat could diffuse downwards. This has a number of effects. First it increases the range of latitudes over which upwelling occurs. Secondly the Ekman transport reduces sea level near the Equator and results in a compensating raising of the density surfaces below. This in turn increases the temperature gradient and aids the downward diffusion of heat.

The main point though is that near the Equator the poleward Ekman transport due to the wind is balanced by a shallow geostrophic inflow. This can be seen in Fig. 2, where at 6°N the Ekman transport has speeds of order 5 $cm\,s^{-1}$ and above 150 m the geostrophic inflow has speeds of around 1 $cm\,s^{-1}$. At the same longitude the core of the NECC lies in the top 150 m, with speeds of only 20-30 $cm\,s^{-1}$. As a result, given the size of the Pacific Ocean, even small secondary flows can have a significant influence on the core waters transported by the NECC.

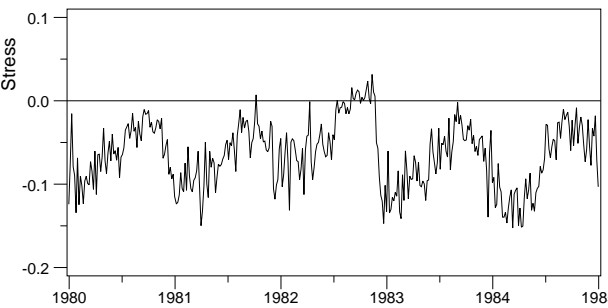

**Figure 10.** Eastward component of wind stress (Pa) at (180°E, 6°N).

### 5.1.1   Ekman transport

Figure 9 shows the eastward component of wind stress at 6°N plotted as a function of longitude and time. It shows a regular pattern each year, the wind stress in the Central Pacific being largest during the northern spring and weakest in summer. The year 1982 is unusual as the stress drops to near zero near the dateline for a large part of the summer and autumn. This is show more clearly in figure 10.

At 6°N, the Coriolis term equals $1.52 \times 10^{-5}$ s$^{-1}$, so if the water density is taken as $1024$ kg m$^{-3}$, then from Eqn.3, the northward transport due to a westward wind stress of 1 Pa is 64.2 m$^2$s$^{-1}$. From Figs. 9 and 10, the westward stress in mid ocean lies around 0.1 Pa in spring dropping to half that value in the autumn. A value of 0.05 Pa will generate a northward Ekman transports 3.2 m$^2$s$^{-1}$, equivalent to 0.36 Sv per degree of longitude. This appears small compared to the NECC transport in a normal year ($\sim$20 Sv, see Fig. 17) but over longitude span of twenty degrees or more it will become significant.

Figure 9 shows that in the autumn of 1982 the zonal wind stress was small over much of the central Pacific. During this period the Ekman transport would have been much less effective in cooling the warm core of the NECC. The figure also shows that the reduction in central Pacific winds at this time is consistent with the arrival of warm water shown in Fig. 6.

### 5.2   Geostrophic inflow

The meridional component of geostrophic transport $V_g$, is related to the zonal gradient of $P$ and the vertical integral of the pressure $p$, by the equations,

$$V_g = (1/(\rho f)) \ \partial P/\partial x,$$
$$P = \int_{-300}^{z_{ssh}} dz \ p(z). \qquad (4)$$

where $z_{ssh}$ is the height of the sea surface, $z$ is depth and $x$ is the co-ordinate in the zonal direction. The lower limit of 300

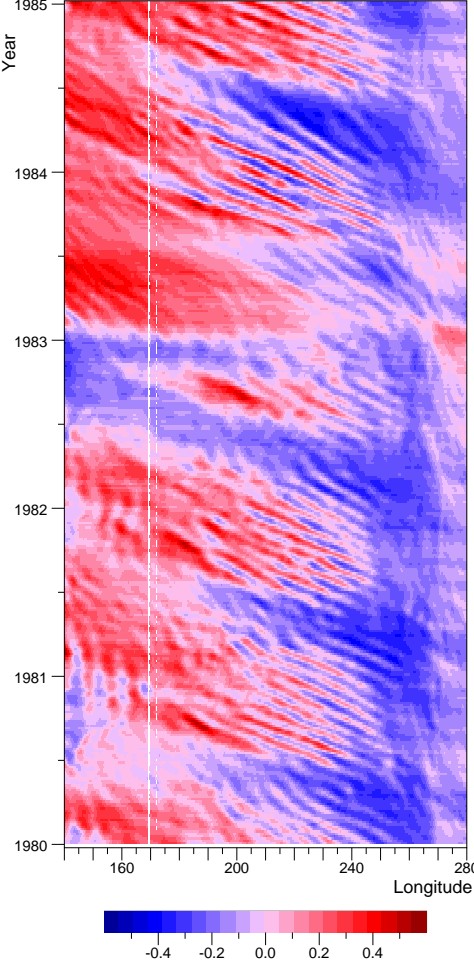

**Figure 11.** Pressure integral ($10^6$ Pa m) defined Eqn. 4 at 6°N (after subtracting a constant equal to the same integral but with a constant density of $1024$ kg m$^{-3}$ and zero surface elevation). Vertical lines are due to shallow topography.

m was chosen because the horizontal gradient of pressure is small at greater depths and the limit is below the normal depth of the NECC.

In Figs. 11 and 12, these variables are plotted as functions of longitude and time at latitude 6°N. As expected, the integrated pressure field is usually greater in the west than in the east, a typical mean gradient between 260°E and 170°E being 0.035 Pa m$^{-1}$. This corresponds to a southward transport of 2.2 m$^2$s$^{-1}$ or 0.25 Sv per degree of longitude.

A second feature that might have been expected near 6°N is the annual Rossby wave (Myers, 1979) which shows up in the integrated pressure field. This has a minimum which starts at the eastern boundary each northern winter and which reaches the western Pacific in the following late summer and autumn. In most years the wave tends to die out west of

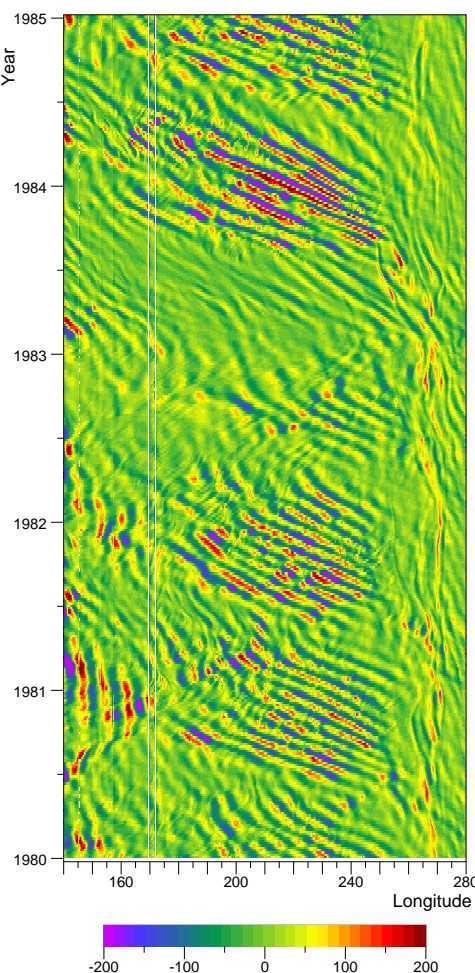

**Figure 12.** Northward component of geostrophic transport ($\mathrm{m}^2\mathrm{s}^{-1}$), defined by Eqn. 4, at 6°N.

200°E, but in 1982, during the development of the El Niño, this does not happen.

In this year there is also a lowering of sea level close to the western boundary, similar to an event seen in 1980. Towards the end of the year there is a second rapid reduction in sea level which affects the western and central Pacific. As discussed later there are similar drops in sea level which occurs at the equator at the same time.

The Rossby wave and the other sea level changes are significant in that they are large enough to reduce and change the sign of the east-west pressure gradient. Thus, especially in 1982, they can significantly affect the flushing of the NECC by the geostrophic inflow.

As in Fig. 6, Figs. 11 and 12 show short wave features which appear to be the result of tropical instability waves. Fig. 12 shows that the meridional transport due to the features can reach values of over 20 $\mathrm{m}^2\mathrm{s}^{-1}$ or 5.6 Sv/degree, suffi-

cient to have a significant effect on the core of the NECC. This aspect is discussed further in the following section.

### 5.3 Tropical instability waves

Tropical instability waves are wave motions observed north and south of the Equator in the Pacific and Atlantic Oceans. They show up most clearly in the surface temperature field as fronts between the cooler equatorial waters and warmer waters to the north and south. In the Pacific they are most noticeable in the eastern Pacific in the late northern summer and autumn.

Understanding of the waves has come primarily through numerical model studies. Philander (1978) used a two layer model and showed that the waves growth was due primarily to a barotropic instability resulting from the strong shear between the Equatorial Current and the North Equatorial Counter Current. Cox (1980), using a multi-layer model, confirmed the importance of barotropic instability but also found that baroclinic instability was involved when the amplitude became large.

However this picture was not supported by the study of Luther and Johnson (1990). They analysed observations made during the Hawaii-to-Tahiti shuttle experiment and found that the main instability lay just south of the Equator and was due to the shear between the Equatorial Undercurrent and the South Equatorial Current. Also unlike Philander and Cox they found an instability between the Equatorial Current and the NECC in the northern winter and a baroclinic instability of the NECC during the Northern spring.

These inconsistencies have never been properly explained, but later studies both observational and numerical (Menkes et al., 2006, see the Menkes paper for more references) support the earlier analysis of Philander and Cox.

Figure 13 shows the model surface temperature fields for late September in 1981 and 1982. The first shows a series of well developed of tropical equatorial waves just north of the Equator starting near 250°E and extending west to beyond 210°E. The corresponding velocity field shows a series of oval anti-cyclonic eddies with an west-east width of about 10 degrees with southern and northern limits at approximately 1.5°N the 7.5°N. The eddies tend to be confined to the top 300 m, the 28°C isotherm which is at a depth of $\sim 20$ m at the equator dropping to around 200 m in the centre of each eddy. Below 200 m the eddy signature drops off rapidly, so although there is some displacement of the isotherms near 500 m the isotherm displacements are very small below that depth.

The eddies are affected by the tropical cell. As a result at a depth of 30 m, maximum northward velocities near 5 °N, are $\sim 1\ \mathrm{ms}^{-1}$ and maximum southward velocities $\sim 0.6\ \mathrm{ms}^{-1}$. In contrast at 108 m, maximum northward velocities are $\sim 0.85\ \mathrm{ms}^{-1}$ and maximum southward velocities $\sim 1\ \mathrm{ms}^{-1}$.

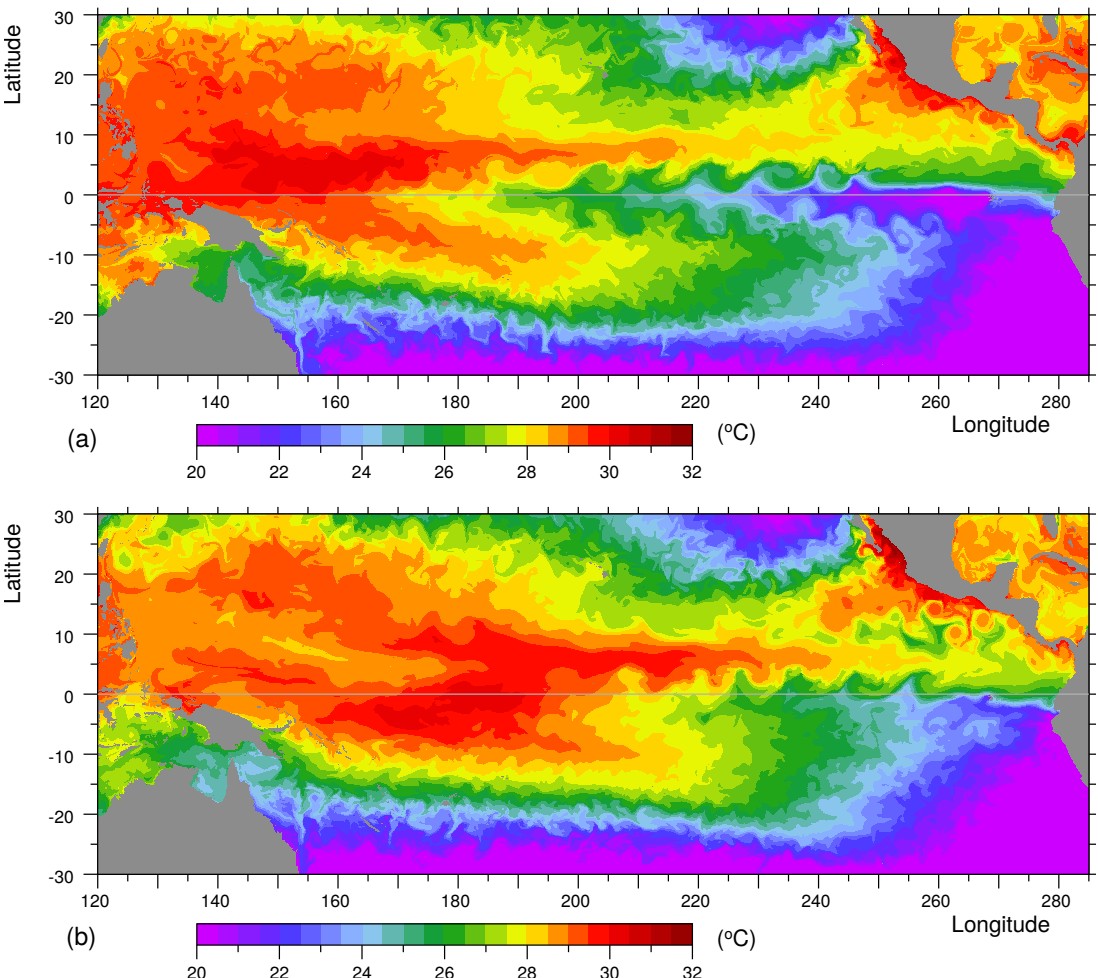

**Figure 13.** Surface temperature (°C) from the model in late September (a) 1981 and (b) 1982. (Values below 20.5°C combined).

### 5.3.1   TIW variability

Given the potential impact of tropical instability waves on the NECC it is useful to have a measure of how their ability to advect water north or south changes with time. This may be achieved by estimating the r.m.s. variance of the northward velocity about its mean value.

Let $V_{300}$ be the northward transport in the top 300 m of the ocean,

$$V_{300} \quad = \quad \int dz \, v. \tag{5}$$

$\bar{V}$, its value over a range of longitudes and $V_{rms}$, the r.m.s. variance, are then given by,

$$\bar{V} \quad = \quad H(V_{300}),$$
$$V_{rms} \quad = \quad H(|(V_{300} - \bar{V})|). \tag{6}$$

where $H()$ is a Hann smoothing filter with a width of 20°of longitude.

The result at 6°N is shown in Fig 14. In most years the r.m.s. transport after smoothing has values around $30 \text{ m}^2\text{s}^{-1}$, consistent with the peak values discussed previously. However what is very significant is the region of very low variability that starts in the west, in mid-1982 and which moves across the Pacific during the latter part of the year. The variability then stays low for a large fraction of 1983.

As the generation of TIWs is partly associated with the Equatorial Current, it is possible that the low variability results from the reduction in the Equatorial Current as the El Niño develops and the low wind stress region moves east.

Figure 15 plots the strength of the surface Equatorial Current as a function of longitude. The region of reduced activity of tropical instability waves, seen in 1982, fits very closely with the region of reduced and reversed currents at the Equator.

This region of low TIW variability can also be seen in Fig. 13. In September 1981, tropical instability waves are mixing cooler equatorial waters into the NECC between 180°E and

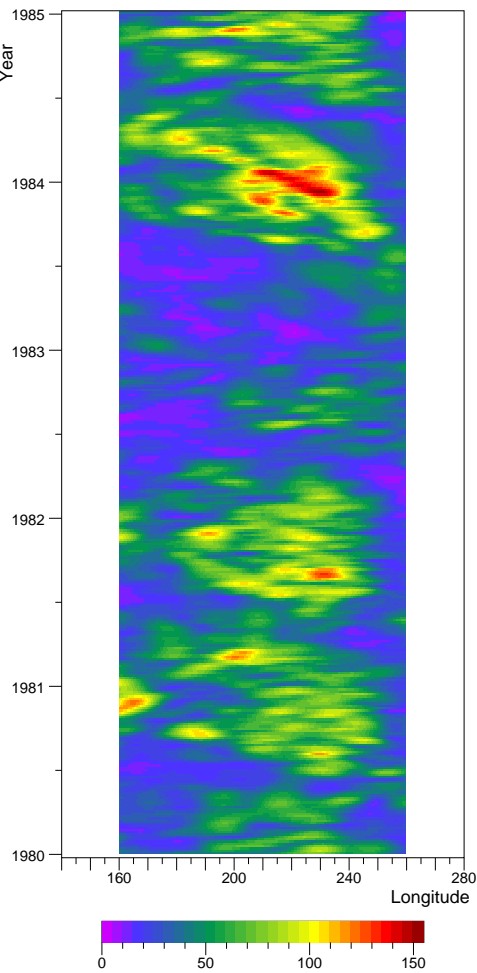

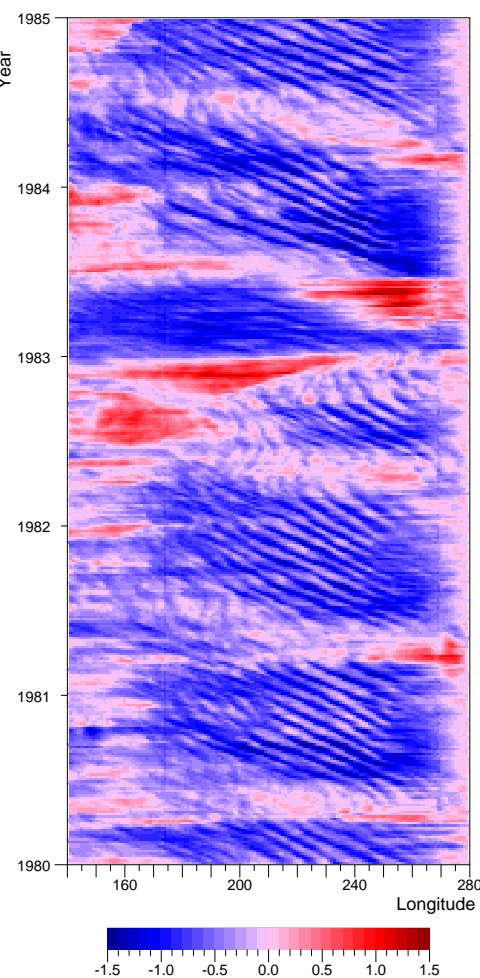

**Figure 14.** The r.m.s. northward transport variability $V_{rms}$ (defined in Eqn.Water 6) along latitude 6°N. Units are $\mathrm{m^2 s^{-1}}$

**Figure 15.** The surface current on the Equator $(\mathrm{m\,s^{-1}})$, averaged between 1°S and 1°N. Negative values correspond to the normal westward flowing Equatorial Current.

200°E. In September 1982 these are not present and the warm core of the NECC is advected much further east before such mixing events occur.

### 5.4 The North Equatorial Counter Current

Any attempt to define the strength of the eastward flowing NECC is complicated by the fact that near the Equator it is often connected to the eastward flowing Equatorial Undercurrent and that at times the wind driven current at the equator may reverse direction. For that reason it is convenient to define the NECC as the region lying between 3°N and 8°N where the surface velocity is eastward and its transport as the integral down to the depth where the velocity first changes sign.

Figure 16, plots the transport defined in this way across 180°E but extended to cover the region from the Equator to 10°N. It shows that when defined in this way the transport

is highly variable. Examination of the velocity field when the eastward flux is zero showed that it was primarily due to oceanic eddies. The large values seen near the Equator arise from the reversal of the surface current at the Equator together with a contribution from the Equatorial Undercurrent.

The year 1982 is seen to be unusual. First the current is continuous, consistent with the reduction in current variability discussed previously. Secondly the peak and average transport in the current appears to increase with the peak value reaching $140\,\mathrm{m^2 s^{-1}}$. Thirdly the latitude of the current core appears to move southward, lying near 5 °N rather than the 7°N that predominates in 1981 and 1982.

Figure 17 plots the total transport between 3°N and 8°N. This shows that the total transport of the NECC averages between 15 and 20 Sv but in 1982 it doubles to between 30 and 40 Sv.

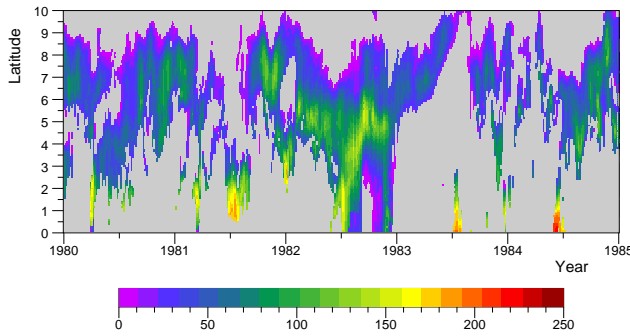

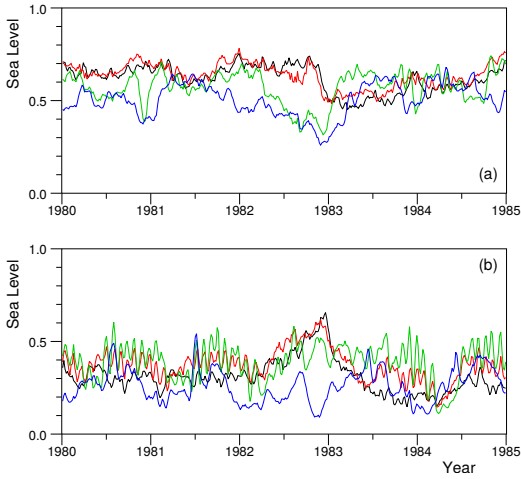

**Figure 16.** Eastward transport $(\mathrm{m^2 s^{-1}})$ of the NECC across 180°E, plotted as a function of latitude and time. The transport here is defined as the integral of the eastward component of velocity from the surface to the first level where it is negative. It is zero if the surface velocity is westward.

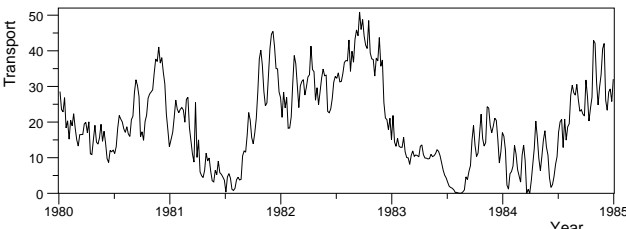

**Figure 18.** Sea surface height (m) at longitudes: (a) 168°E, (b) 230°E and latitudes: (black) the equator, (red) 3°N and (green) 6°N and (blue) 9°N.

**Figure 17.** Total transport (Sv) between 3°N and 8°N of the NECC across 180°E.

## 6 Differences in El Niño years

The results presented so far show that the NECC can at times transport large amounts of water with temperatures above 28°C eastwards across the Pacific. They also show that in
most years this does not occur because tropical instability waves, the Ekman transport and the geostrophic inflow combine to dilute the warm core of the NECC with cooler water from the north and south.

However in an El Niño year, once the region of low wind
stress has started moving eastwards, the strength of these processes is reduced in the ocean to its north and west. As a result the core of the NECC passing the eastern boundary of the low wind region is much warmer than normal and as it continues eastwards it has the potential to trigger new
episodes of deep atmospheric convection. As a result the region of deep atmospheric convection may progress steadily eastwards.

This poses the question "Why does an El Niño not occur every year?", or given that the processes that start an El Niño
has not been discussed "Why is not every El Niño a strong El Niño like the one in 1982-1983?".

One possibility, originally proposed by Wyrtki (1974) and supported by the results of the last section, is that the year

to year differences are, in part, a result of changes in the strength of the NECC. For this reason the next sections con- 25 sider the year to year differences in more detail.

### 6.1 Wyrtki's NECC estimate based on sea levels

Wyrtki (1974) estimated changes in the strength of the NECC from sea level measurements made at Kiritimati (Christmas Island, 01°52'N 157°24'W) on the Equatorial Ridge and 30 Kwajalein Atoll (8°43'N 167°44'E) in the Counter Current Trough. He found that the height difference was largest and the NECC presumably strongest during the El Niños of 1957-1958, 1963-1964 and 1967-1968.

The model data was analysed in the same way and it was 35 found that the SSH difference between 3°N and 9°N correlated with the average surface currents between those latitudes. In particular they both showed significant increases during the same periods in the autumns of 1982 and 1997, when the strong El Niños shown in Fig. 5 were developing. 40

Wyrtki's analysis showed that the change in NECC strength was due primarily to the lowering of SSH in the Counter Current Trough. As shown in Fig. 18a, at 168°E, the longitude of Kiritimati, the models results agree with this.

They show a reduced sea level in the trough, at both 6°N 45 and 9°N, during the second half of 1982 but a roughly constant sea level near the equator during the same time period. At other times, sea level at 3°N is usually slightly above that at the Equator, a result of the Equatorial Trough that develops when the westward flowing Equatorial Current is present. 50

Further east at 230°E (130°W) (Fig. 18b) sea level differences between latitudes are generally smaller and there is a strong annual signal, especially at 6°N. In the second half of 1982, there is again a large sea level difference between the Equator and 9°N. However at this longitude the main slope 55

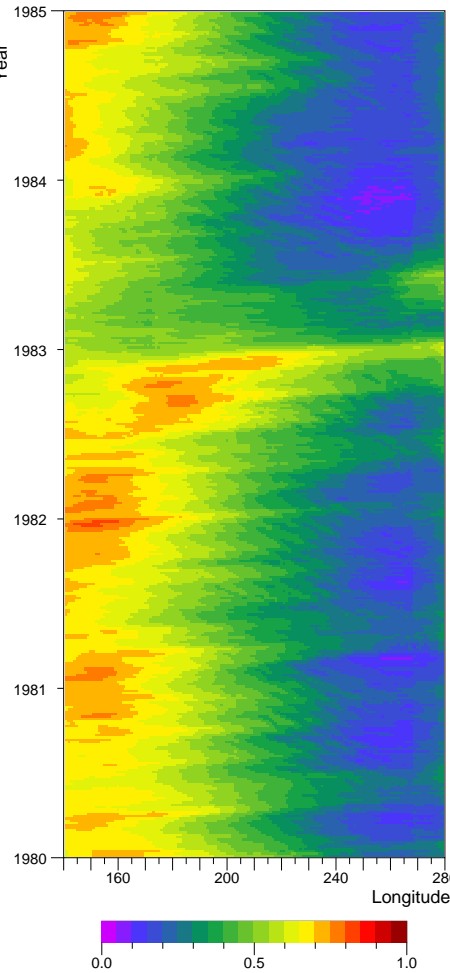

**Figure 19.** The model SSH (m) at the Equator as a function of time and longitude.

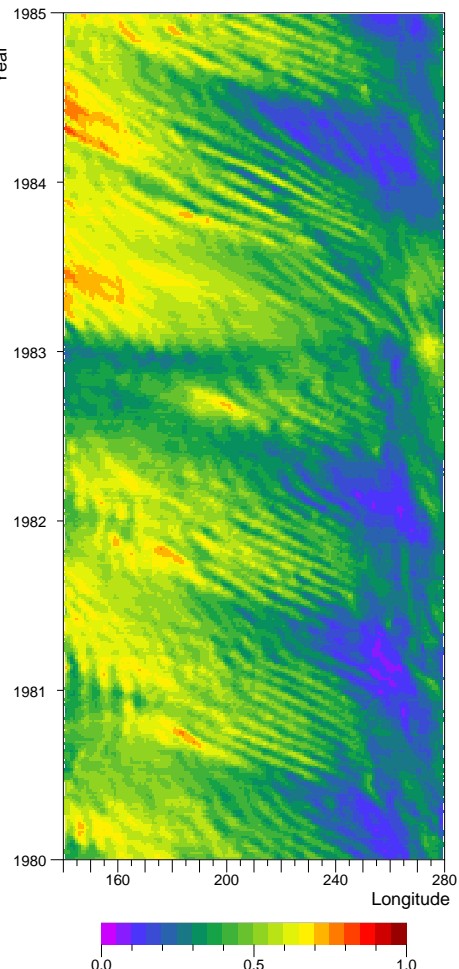

**Figure 20.** The model SSH (m) at 6°N as a function of time and longitude.

lies further north between 6°N and 9°N. It also arises primarily from an increase in sea level near the Equator, the sea level in the trough at 9°N remaining relatively constant.

Thus the model agrees with Wyrtki's result for Kiritimati
but it also indicates that the full picture is much more complex. To understand more it is convenient to investigate the changes in sea level with both longitude and time at each latitude.

## 6.2 The annual wave and other processes

Figures 19 to 21 show the sea level plotted as a function of longitude and time at the Equator, at 6°N and at 9°N.

Starting with the equator, the figure shows that, except during the 1982-83 El Niño event the east-west slope remains relatively constant. Eastward traveling equatorial Kelvin
waves occur at regular intervals which, east of their gener-

ation region, produce increases in sea level which return to normal after the wave has passed.

In this figure the El Niño event starts in the middle of 1982 when the sea level in the west drops and the region of maximum sea level moves to approximately 190°E. The initial 20 movement may be associated with a Kelvin wave, but the maximum then remains fixed, despite further Kelvin waves, until near the end of 1982 when sea level drops rapidly all along the equator. This collapse is certainly associated with a Kelvin wave. 25

Following the collapse, sea level stays low throughout 1983, recovers slightly in 1984 and only returns to normal at the start of 1985.

At 9°N, sea level again shows a mean east-west slope, but at large scales it is highly variables, the east-west dif- 30 ferences being largest in the middle of 1981, 1983 and 1984 and smallest at the end of 1982. The latter occurring around the period when sea level dropped along the Equator. How-

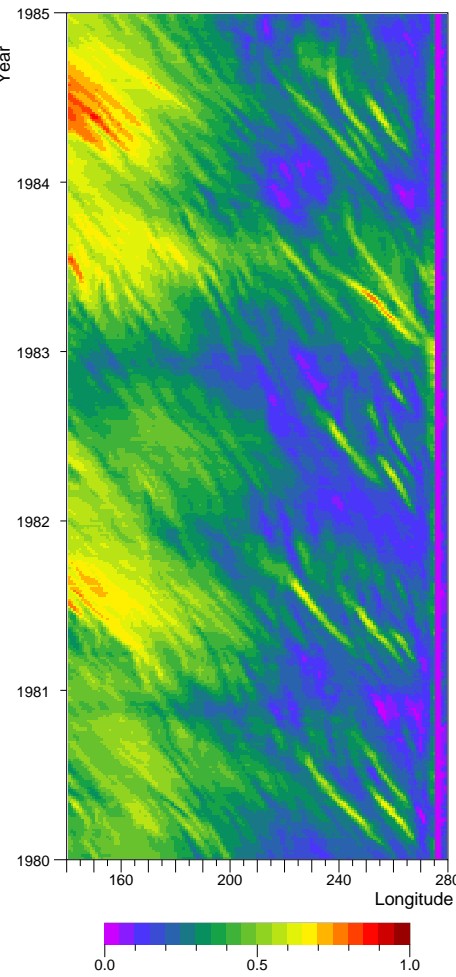

**Figure 21.** The model SSH (m) at 9°N as a function of time and longitude.

ever, unlike the Equator, sea level rapidly recovers in 1983 to a value in the west even higher than in 1981.

Sea level at 9°N also shows short period and short wavelength Rossby wave like features moving westward at all times. The features may be partly due to tropical instability waves, but the region is also affected by eddies along the edge of the North Equatorial Current.

The annual signal at 9°N is strong and to first order appears to consist of two main components. The first is a change independent of longitude which has its maximum in the middle of each year. The second is a set of westward traveling waves, an example of which is the minimum in sea level that starts at the eastern boundary in the autumn of 1981 and which reaches 200°E at the end of 1982.

At 6°N, sea level also shows an annual variation, but here the signal appears to be dominated by the westward traveling annual wave. Like the wave at 9°N this starts at the eastern boundary late in the year. It reaches 260°E (100°W), the ap-

proximate longitude of the Galapagos Islands, in the northern spring where it is associated with a minimum in sea level. It then moves westward more rapidly, the leading edge reaching the western boundary in mid year and the trailing edge arriving before the end the year.

The propagation of the 1982 minimum in sea level at 6°N appears to be unusual in that at 230°E minimum is similar to the value in 1981 but in the region west of 180°E the minimum is much lower.

As shown in Fig. 18, at 168°E the passage of the wave results in the sea level at 6°N being similar to that at 9°N at a time when sea level at the Equator remains high. Thus although the meridional pressure difference across the NECC remains roughly constant, the current is squeezed into a path nearer the Equator, where the Coriolis term is smaller. As the current is in geostrophic balance, its transport per unit depth must increase.

Thus a stronger than normal annual Rossby wave will move the core of the NECC towards the equator, increasing the speed of the current and the flux of warm water to the east. The increased speed will help to reduce the effect of tropical instability waves and the other mechanisms on the core temperature of the NECC. If the core temperature is high enough and the flux large enough this may then trigger new episodes of deep atmospheric convection further east in the Pacific.

## 7   Development of the El Niño during 1982

The discussion so far has concentrated on individual physical processes with only limited discussion of the overall development of the El Niño. To give more context, the following sections briefly discuss some of the other events that occurred in the equatorial Pacific during 1982 and how these may be connected to the processes discussed above.

### 7.1   29th March

Figure 22 shows fields of sea level and surface temperature together with the surface velocity and wind stress vectors for the 29th March 1982. At this time the minimum in the annual wave at both 6°N and 9°N is still in the eastern Pacific, where it contributes to the minimum in the Counter Current Trough near 240°E. The vector plot shows that he NECC is a weak feature, except between 140°E and 160°E where it runs along the northern flank of a region of maximum sea level.

The figure also shows the Equatorial Current in the central Pacific with sea level ridges to north and south on which can be seen maxima due to tropical instability waves. The warmest temperatures on the equator are found north-east of New Guinea and this is also the region where sea level along the Equator is highest.

The figure shows winds flowing in a south-east direction along the north cost of New Guinea. Such winds often oc-

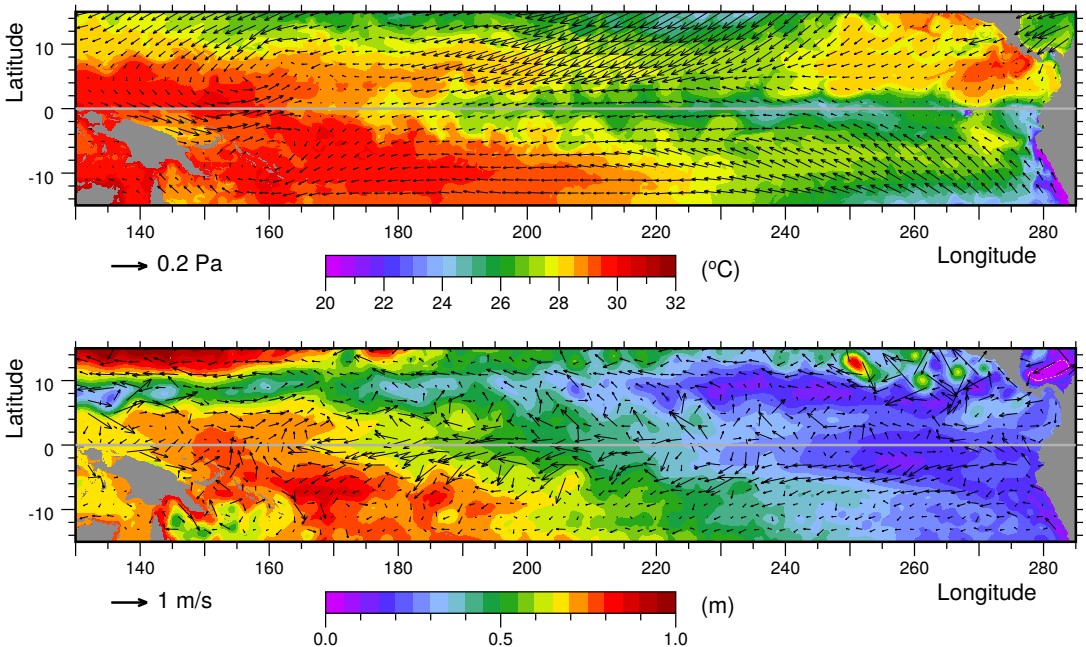

**Figure 22.** Upper: Surface temperature and wind stress vectors. Lower: Sea level (SSH) and velocity vectors from the 29th March 1982 archive dataset. Each archive dataset contains averages over the previous 5 days of the model run.

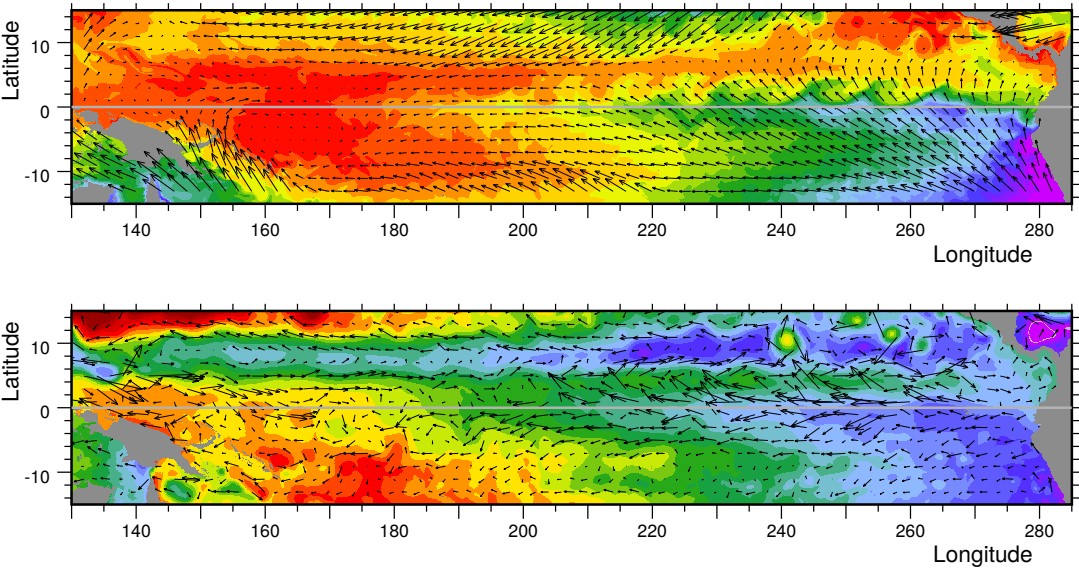

**Figure 23.** Upper: Surface temperature and wind stress vectors. Lower: SSH and velocity vectors from the 29th June 1982 archive dataset. Colours and vector scales as in Fig. 22.

cur early in the year when there is strong convection in the South Pacific Convergence Zone. In this case convection over warm water appears to have generated cyclones both north and south of the Equator generating, for just one five day averaging period, an extended region of westerly winds along the Equator.

In the ocean the Equatorial Current was still present early in the month but has now disappeared. It is not re-established but instead, during May and June, there are periods with Reversed Equatorial Current between 150°E and 170°E.

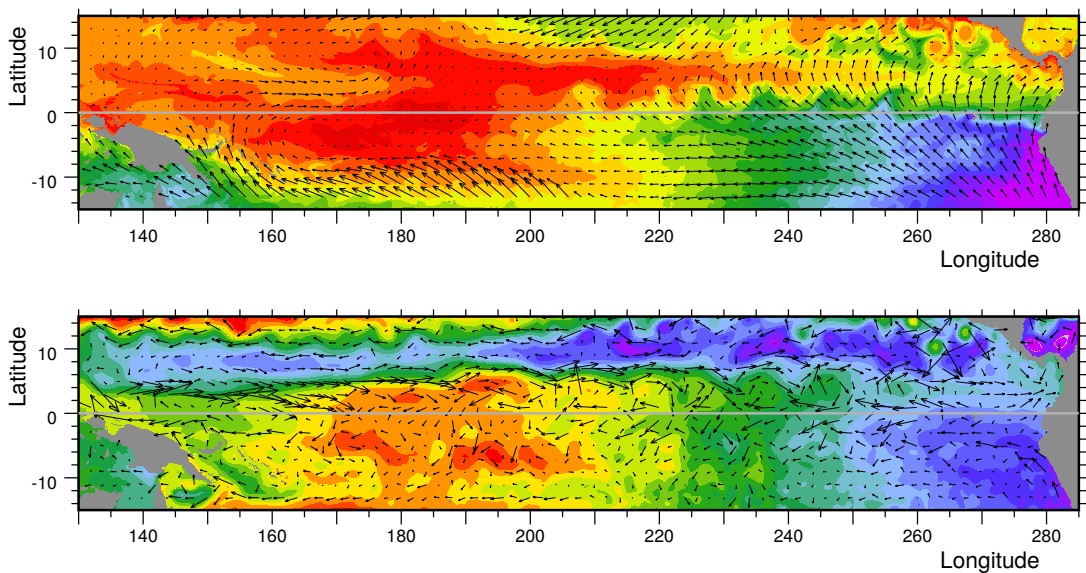

**Figure 24.** Upper: Surface temperature and wind stress vectors. Lower: SSH and velocity vectors from the 27th September 1982 archive dataset. Colours and vector scales as in Fig. 22.

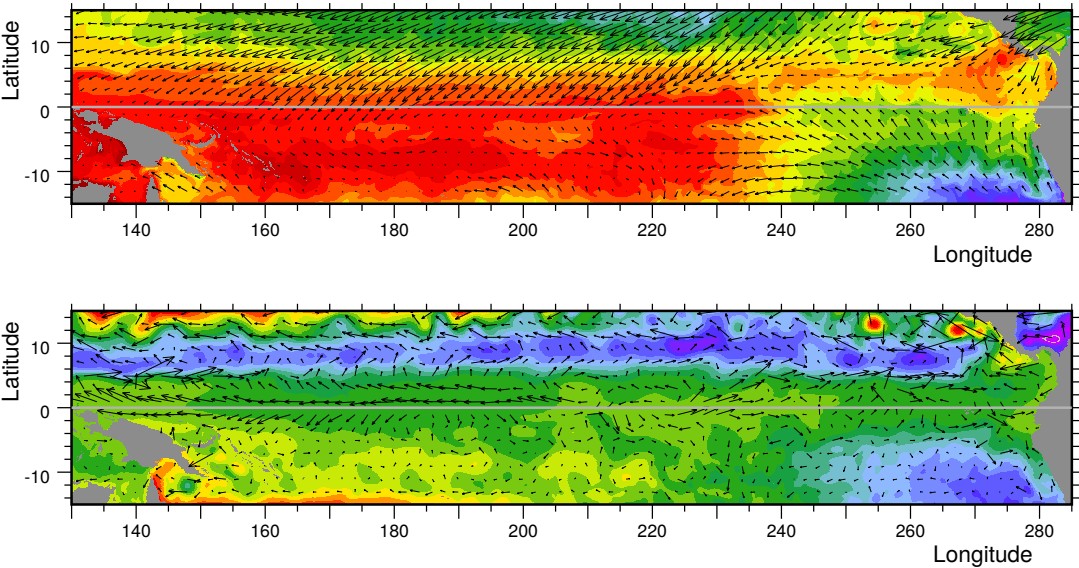

**Figure 25.** Upper: Surface temperature and wind stress vectors. Lower: SSH and velocity vectors from the 31st December 1982 archive dataset. Colours and vector scales as in Fig. 22.

### 7.2   29th June

By the end of June the annual wave at 6°N has reached the western Pacific and the wave at 9°N has reached 230°E. As a result the Counter Current Trough is deeper and more uniform throughout the central and eastern Pacific. The Equatorial Ridge in the eastern Pacific is also more developed, and

this, together with the changes in the trough result in a much stronger NECC all across the Pacific.

In the west, sea level on the Equator has dropped slightly, but this is the time that in Fig. 19 the maximum sea level is in the process of moving from around 150°E to 190°E (170°W). The region of low winds has started to expand, temperatures have risen, including along the line of the NECC, and the

current is more effective at transporting warm water to the east, beyond the region of low winds.

### 7.3   27th September

This lies in the middle of the time period when the maximum SSH along the Equator lies near 190°E. The temperatures within the region increase with time. The region also spreads north and south on both sides of the Equator. One consequence of this is the region of higher than normal sea level at 20°E, 6°N seen in Fig 20.

By this time the NECC has grown in strength and its path shifts northwards as it crosses the ocean, starting near 4°N and reaching 8°N near 240°E.

This is also a this period when westerly wind bursts develop. These can be seen in Fig 3 and the resulting Kelvin waves in Fig 19. However these occur in the region where the mean wind is now westerly and there is no evidence that the resulting eastward surface current along the Equator in this region is significantly different from that to be expected from the average westerly wind.

### 7.4   31st December

In October and early November 1982 the central and western equatorial Pacific was a region of light westerly winds interspersed by stronger westerly wind bursts. In contrast in the Eastern Pacific, a strong trade wind continued to blow.

In late November the pattern changed and strong westerly winds blew along the equator on either side of the dateline. These result in an eastward flowing surface current which continues until the end of the year (Fig. 25), advecting the warm water patch on the equator towards the east.

By the end of the year, the trade winds are starting to be re-established north of the Equator. In the west the Equatorial Current is reforming and by the 10th January it is again established in the east Pacific. As a result in the following weeks the patch of warm water on the equator moves back westwards.

However, as shown in Fig. 4, this is the time that precipitation is most established in the central Pacific. Precipitation remains high in the central Pacific during the remainder on the Southern Hemisphere summer after which it first moves closer to South America before the high precipitation region is re-established in the western Pacific.

### 8   Particle tracking

A useful alternative view of the processes can be obtained using the Tracmass particle tracking program (de Vries and Döös, 2001). Figure 26 shows the results of seeding the NECC at 200°E in June 1981 and 1982. This is the time when in 1982 the NECC was carrying water warmer than 29°C into the western Pacific.

In 1982 each model grid box along the line shown, with a temperature greater than 29°C, was seeded with a single particle. In 1981 there was no water along the line with this temperature so boxes were seeded where the temperature was greater than 27.8°C.

In 1981, the water was initially carried east but before reaching the far eastern Pacific most particles moved north, where they were carried westward by the North Pacific Subtropical Gyre. The remainder moved south and were carried westward by the Equatorial Current.

By contrast, in 1982 the much warmer water was carried predominantly to the east with a significant quantity reaching as far as the Galapagos islands. Particles were carried to the north and south, but fewer were lost in this way than in 1981.

In two other runs (not shown), particles were seeded in the Niño 1 and 2 regions, which are to the south-east of the Galapagos Islands. The tracking program was then run backwards in time, from early in 1982 and 1983, to determine where the water came from.

In 1982 the water was found to have a local origin, some coming from upwelling regions near the coast. In contrast in 1983 a significant amount came from just north of the Galapagos, apparently displaced by the water entering the region shown in Fig. 26. Observations made in 1981 and late 1982 show a similar movement of warm, low nutrient, surface water southwards across the Equator at this time (Barber and Chavez, 1983).

In a final test, water particles were tracked moving eastward along the Equator. The model showed that in late October 1982, following the reduction and reversal of the winds, the Equatorial Current at 200°E also reversed direction. The region of water warmer than 29°C, shown in Figs. 13 and 24, then started moving eastwards along the Equator.

This ocean was seeded as before and Fig. 27 shows the initial seeding line and the later particle positions. The particles are seen to move eastward but they do not progress far. By the end of the year none of the particles have passed 240°E and many have turned back westward.

### 9   The 1997-1998 El Niño

As a check that the above results were not unique, the analysis was repeated for the strong 1997-1998 El Niño. Key results are shown in Fig. 6 and in Figs. 28 to 35.

The temperature contours in Fig. 6b indicate that the 1997-1998 El Niño started much earlier in the year than that of 1982-1983. During the northern spring months the region with SST values above 28°C, extended eastwards until mid-year when it had reached 230°E, the limit being similar to that of the warm pool events of 1987-1988 and 1991-1992 (see Fig. 5).

In 1997, the temperature contours then briefly retreat westwards before being overtaken by the main El Niño event

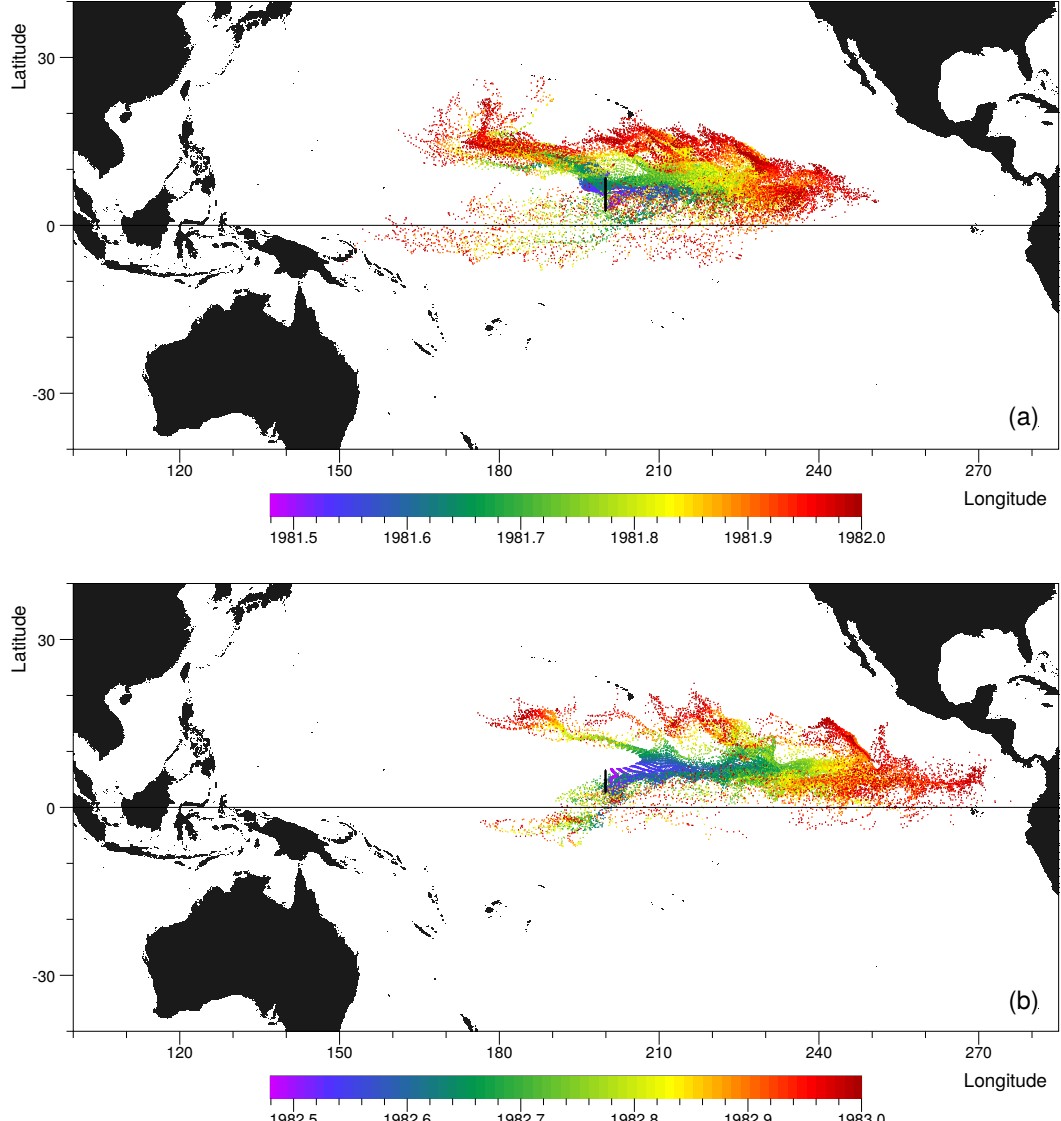

**Figure 26.** Water particle positions plotted every 5 days, starting from the 24th June (a) 1981 and (b) 1982 and running to the end of the year. The date is denoted by the colour of each dot. For the initial state, one particle was placed at the centre of each model grid cell lying along the black line, having a water temperature of greater than (a) 27.8°C and (b) 29°C, there being no water with a temperature greater than 29°C along the line in 1981.

which carries water with a temperature greater than 28°C to the eastern boundary.

Figure 28, shows the flux of water with temperatures greater than 28°C across 180°E, 210°E and 240°E. The first corresponds to a longitude well withing the warm pool event, the second to a longitude near its the eastern limit and the third to a longitude beyond the limit.

At 180°E the transport of warm water in 1995 and 1996 by the NECC is seen to be greater than that in the year preceding the 1982-1983 El Niño. This may indicate that the western Pacific was warmer during the later period. The NECC continues with a similar flux of warm water during early 1997, but in early spring, at about the time the temperature front of

Fig 6 reaches 180°E, there is a significant flux of warm water in the equatorial band and this continues until the end of the year.

At 210°E in the equatorial band there is a single pulse of warm water in mid-year, when the warm water of Fig. 6 reaches this longitude, but for the rest of the year the main transport is at the latitudes of the NECC. At the end of 1997, as in 1982, there is again a short pulse of warm water in the equatorial band.

At 240°E, during 1997, warm water is only advected by the NECC, and as shown in both the figures for 180°E and 210°E, this is associated with a movement of the NECC towards the Equator.

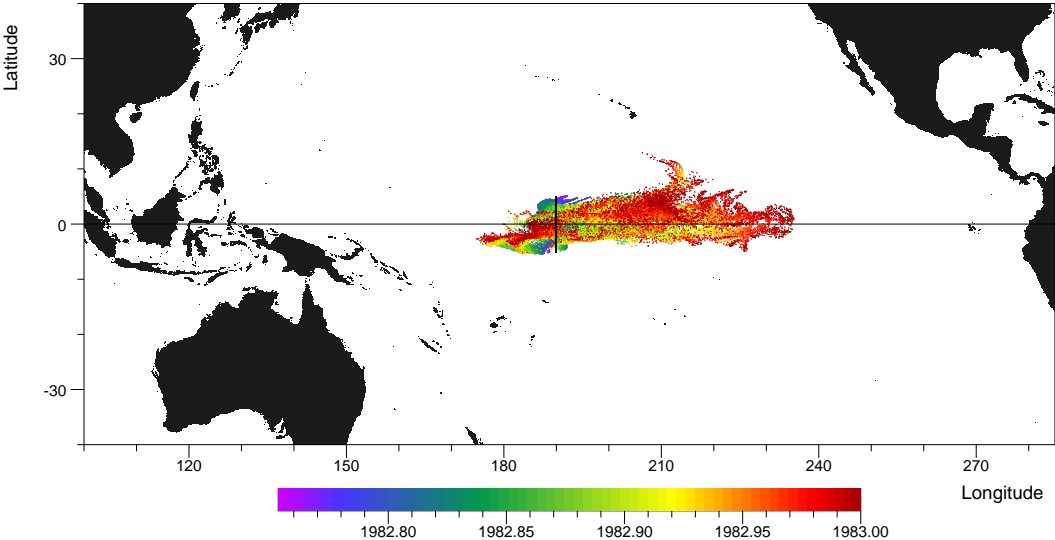

**Figure 27.** Water particle positions plotted every 5 days between 2nd October 1982 and the end of the year, the date being denoted by the colour of each dots. For the initial state, one particle was placed at the centre of each model grid cell lying along the black line having a water temperature of greater than 29°C.

Overall the results indicate that the strong 1997-1998 El Niño was different in that it developed from a warm pool event whose maximum occurred around mid-year. However the NECC was again involved in the second half of the year, transporting warm water eastwards well beyond the limit of the warm pool event and eventually into the eastern Pacific.

### 9.1 Dilution Processes during 1997-1998

Figure 29a shows the eastward wind stress at 6°N, responsible for the Ekman transport contribution to the dilution of the NECC at that latitude. During the early part of 1997, winds in the central Pacific are weaker than in the corresponding period of 1982 and in the west the winds are mainly either near zero or westerlies. As in 1982-1983 the region of low winds lies well to the west of the warm water boundary seen in Fig. 6b.

During the early Autumn there is a second period of stronger westerlies in the west but these are weaker than in 1982. Winds remain low or westerly across the whole of the Pacific, until late in the year when the normal pattern of easterly winds starts to return.

The pressure integral term (Fig. 29b), shows the influence of a stronger than normal annual Rossby wave in 1997 similar to the wave that occurred in 1982. The integral also has values which are much lower than normal along the western boundary, during the autumn of 1997, and all across the ocean towards the end of the year. As in 1982, the changes in the west and at the end of the year, appear correlated with changes in sea level along the Equator (Fig. 31) occuring at the same time.

The variability of the pressure gradient (Fig. 29c), shows that the tropical instability waves are reduced in intensity during the development of the 1997-1998 El Niño, again as they were in 1982-1983. The region of reduced variability starts in the west in early spring 1997 and gradually extends eastwards until the end of the year. In the following year the region slowly retreats and is replaced, again as before, by a stronger than normal set of waves that develop in the central and eastern Pacific.

The plot by Moum et al. (2009, Fig. 4) based on mooring data from the Equator at 220°E (140°W), shows a similar reduction in the strength of the tropical instability waves during the second half of 1997 and the early months of 1998.

### 9.2 Sea Level during 1997-1998

Figure 30 shows the sea surface temperature field in late September 1997. Although the El Niño started in a different way, at this stage the temperature field is very similar to that from the 1982-1983 event. This will have been partly due to the reduction in the dilution processes discussed in the last section. However following the analysis of 1982-1983 El Niño, it may also be due to changes in the strength of the NECC resulting from changes in sea level at the Equator and at latitudes close to the NECC.

Figure 31 shows sea levels at the Equator, 6°N and 9°N during the period 1995-2000. In most years, sea level at the Equator shows the expected increase from east to west due to the trade winds. Sometimes, as in early 1996, there is a small reversal in slope close to the western boundary. When this does occur it is usually near the turn of the year when winds on there are often westerly (Figs. 3 and 32). Equato-

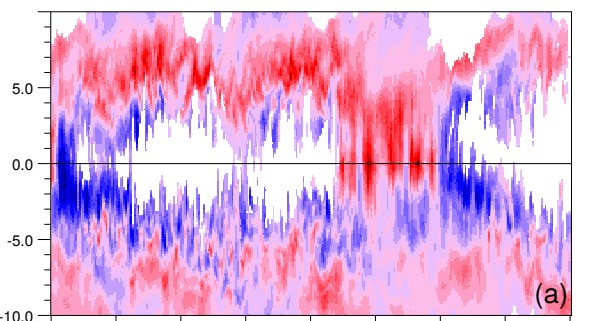

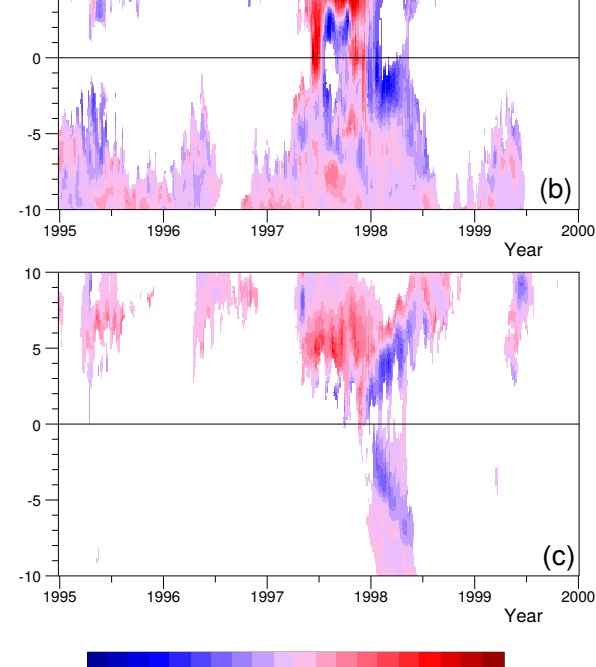

**Figure 28.** Vertically integrated flux of water $(\mathrm{m^2 s^{-1}})$ with temperature (a) greater than 28°C crossing longitude 180°E, (b) greater than 28°C crossing 210°E, (c) greater than 28°C crossing 240°E. The figure is blank where the flux is zero.

rial Kelvin waves are also seen in the sea level figure but as before they to not observed in the surface temperature plot (Fig. 6b).

In early 1997, the maximum in sea level moved away from the western boundary. The behaviour is similar to the 1982 event but this time the maximum dies away and there is a period of reduced east-west slope, with lower than normal sea levels in the west and higher than normal sea levels in the eastern Pacific.

In the autumn, a second region of high sea levels develops near 190°E and, as in 1982, it remains in approximately the same position until late in the year. It then moves slightly eastwards before sea level again falls rapidly all along the Equator.

Sea level at 6°N also has strong similarities with the earlier period, the annual Rossby wave reaching the western Pacific in the second half of 1997. There is also a reduction of equatorial sea level near the western boundary at this time which spreads eastward to extend along much of the Equator by the end of the year. This similar to the change in the pressure integral discussed in the previous section. A plot of the difference in sea level between 6°and the Equator shows that this is approximately constant in the spreading region as the sea levels drop.

At 9°N, sea level also shows a reduction in east-west slope during late 1997 (and early 1998). In the west sea level is higher than at 6°N, implying a westward current probably due to a southward extension of the North Pacific Gyre. In the central and eastern Pacific sea level is lower than at 6°N, this implying that the path of eastward flowing NECC has moved further north here.

## 10    Developments during 1997

### 10.1    16th March

Figure 33 is included partly to illustrate the cross equatorial wind flows that often occur near the beginning and end of each year in the western Pacific. The winds cross the Equator north of New Guinea and are responsible for many of the positive values seen west of the dateline in plots of the eastward wind stress along the Equator (Figs. 3 and 32).

After crossing the Equator the winds continue towards the South Pacific Convergence Zone, where deep atmospheric convection events are expected to be strongest at this time of year. Westerly winds along the Equator may also be produced by cyclones close to Indonesia and, once the warm water front has moved further east, as a result of cyclones that develop north or south of the Equator over the warm water.

In mid-March the strong westerly wind event lasted for almost 20 days (Fig. 32) and as seen in Fig. 33 this resulted in a strong current along the Equator advecting a surface water mass with temperatures up to 30°C.

The figure corresponds to the period when the warm pool event was developing. At this time the NECC is relatively weak. It is advecting some water warmer than 28°C to the east, but tropical instability waves are well developed near 180°E, and these are rapidly mixing away warm water from the core of the current.

### 10.2    29th June

Figure 34 corresponds to the end of the western and central Pacific warm pool event and the start of the eastern Pacific

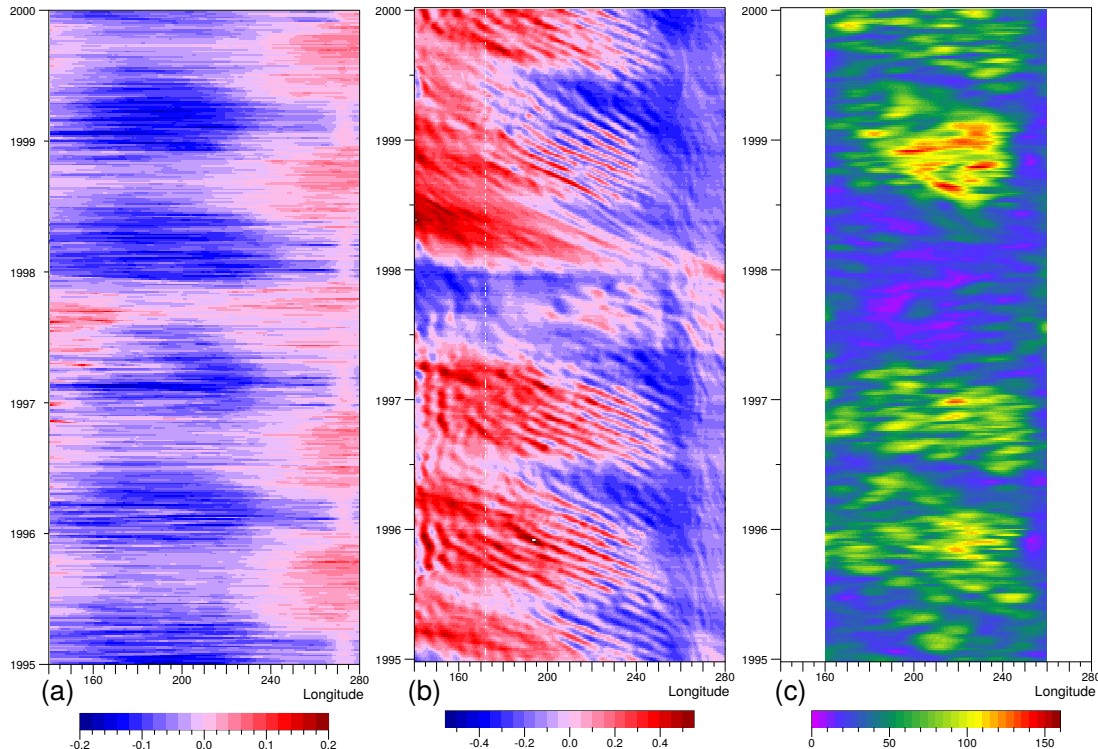

**Figure 29.** Values at 6°N, during the period 1995 to 2000, of (a) the eastward component of the wind stress (Pa), (b) the pressure integral ($10^6$ Pa m) of Eqn. 4 at 300 m, (c) the r.m.s. northward transport variability $V_{rms}$ (m$^2$s$^{-1}$).

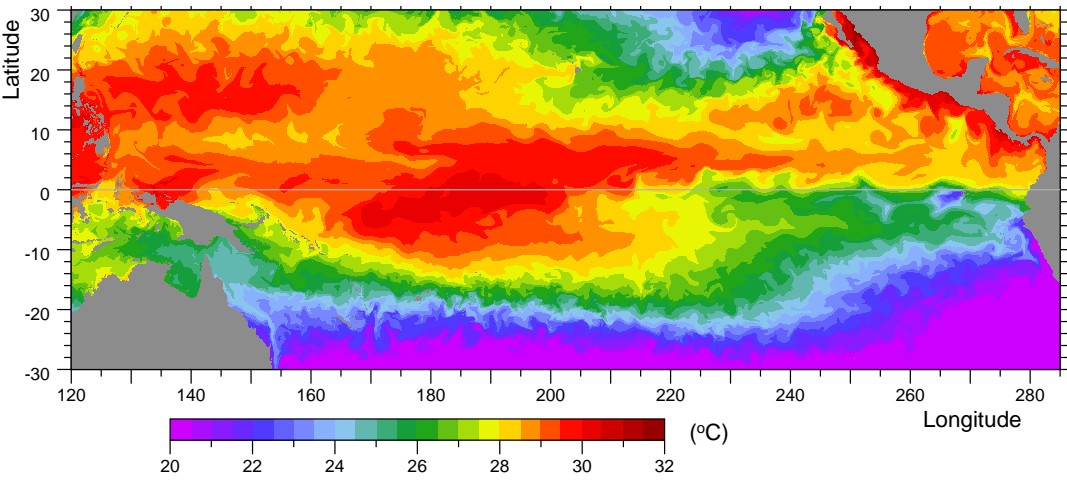

**Figure 30.** Surface temperature (°C) from the model in late September 1997 (Values below 20.5°C combined).

El Niño. On the Equator the winds are predominantly from the east, but earlier in the month a strong Reverse Equatorial Current developed in the central Pacific which appeared to be closely linked with the NECC. The current is still present here and is associated with a ridge of high sea level along the Equator. Sea surface temperatures of greater than 29.5°C

are found in the central Pacific along the Equator and to the north.

At 8°N, the Counter Current Trough is well develop resulting in a well developed NECC over most of the width of the Pacific. At the longitudes where there is a ridge along the Equator, the two currents appear to combine generating a single broad Reverse Equatorial Current.

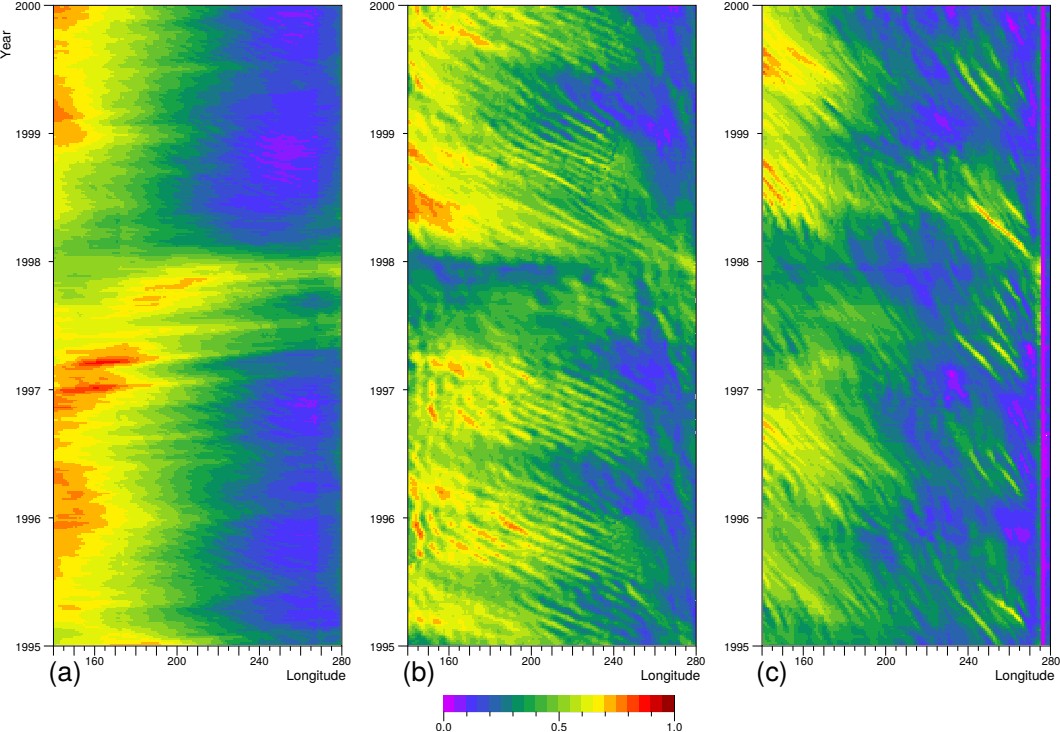

**Figure 31.** Sea surface height (m) during the period 1995 to 2000 at (a) the Equator,(b) 6°N, (c) 9°N.

### 10.3   27th September

The overall picture (Fig 35) is similar to that of September 1982. The Counter Current Trough remains well developed and there is a strong NECC carrying warm water to the far western Pacific. The eastward flowing Reverse Equatorial Current has disappeared and in the eastern Pacific the westward flowing Equatorial Current has returned. There is some upwelling of cooler water near the Galapagos but, compared with the same time in 1982, tropical instability waves are less developed.

On the Equator, the region of low winds extends to 210°E. Near the dateline, where sea surface temperatures are high, the wind stress vectors show convergence on the Equator. This may be connected with continuing deep atmospheric convection in the region.

### 11   Discussion

This paper is the result of a preliminary analysis of archived data from an early run of a high resolution global ocean model. A previous comparison with observations from the equatorial Pacific indicated that the model behaved well and so provides some measure of confidence in the present results.

The analysis shows that during the development of the strong 1982-1983 El Niño, the North Equatorial Counter Current dominated the transport of water with temperatures greater than 28°. This was also true during the development of the strong 1997-1998 El Niño at longitudes east of 130°E. The path of the NECC lies close to the latitude of the Intertropical Convergence Zone so the atmosphere is likely to be very sensitive to warm water carried eastwards by the NECC.

The analysis also showed that the movement of warm water along the equatorial band during the two strong El Niños was very different from the warm pool events of 1987-1988 and 1992-1993, and the similar event of early 1997. In the strong events warm water spreads rapidly eastwards across the Pacific after which it equally rapidly retreats. The warm pool events are more incremental, the westward extend usually extending slowly from one year to the next to a maximum near 230°E, after which there is a relatively small retreat.

During the growth of the strong El Niños, and also during the warm pool event of early 1997, the core temperature of the NECC is higher than usual. This is associated with a reduction at 6°N of the Ekman transport, geostrophic inflow and tropical instability waves, all of which can remove warm water from the core of the NECC and replace it with cooler water from north and south.

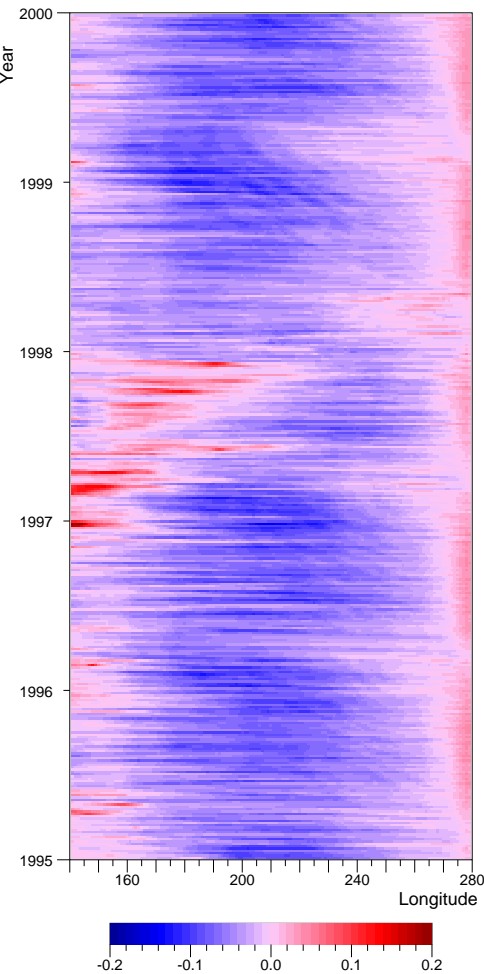

**Figure 32.** Eastward component of wind stress (Pa) at the Equator between 140°E and 280°E (80°W).

The reduction in the Ekman transport is associated with reduced winds in regions where deep atmospheric convection appears to have moved out over the ocean. The reduction in the strength of the tropical instability waves, potentially the most important process, is associated with a reduction in the strength of the Equatorial Current. This can also result from reduced easterly winds. The reduction in geostrophic inflow may partially result from the reduction in Ekman transport but the model results also show that it is connected with the passage of the annual Rossby wave.

During the growth of the strong El Niños, the NECC is observed to move nearer the equator and become stronger. The model results indicate that this is also a result of the passage of a stronger than normal annual Rossby wave. In the west the wave deepens the Counter Current Trough thus increasing the strength of the NECC. In the central Pacific the wave moves the northern boundary of the current southwards but produces little change in the north-south pressure difference

across the current. As the Coriolis term drops to zero at the equator, this will inevitably increase the speed of the current.

In both of the two strong El Niños and in the warm pool event of early 1997, sea level on the Equator developed a maximum in mid-ocean. Once this had formed, its position remained relatively fixed despite the continuing eastward extension of the pool of warm water. Near the end of the event the maximum moved slightly eastwards before sea level dropped all along the Equator. The reason for the behaviour is not understood but the maximum was usually associated with the warmest patch of water lying on the Equator.

The forcing fields show that periods with strong westerly winds occurred on the Equator during the development of both the two strong El Niños studied and the warm pool event. North of New Guinea, on either side of New Year, this was often due to a cross equatorial airflow towards the South Pacific Convergence Zone.

At other times strong westerlies and associated cyclones were only found above regions where the water temperature was already above 28°C and deep atmospheric convection is likely to have occurred. The strong westerlies on the Equator did drive a Reverse Equatorial Current but this was confined primarily to the region of warm ocean. The westerly winds also generated equatorial Kelvin waves but there is no evidence that these caused a significant extension of the warm water region.

## 11.1 Ocean Mechanisms

The results highlight two oceanic mechanisms that are important during the development of a strong El Niño.

The first is the Rossby wave mechanism that increases the speed of the North Equatorial Counter Current. In the west the annual Rossby wave deepens the Counter Current Trough. In the central Pacific it moves the NECC closer to the Equator into a region where the Coriolis term is smaller.

On the basis of the present model results, the timing of the strong El Niños is almost certainly due to the the annual Rossby waves, the arrival of the wave at 6°N in the western Pacific in mid-year lowering sea level and triggering the increased transport by the NECC. As it crosses the Pacific the flow of warm water is aided slower moving Rossby waves until it arrives in the far eastern Pacific around the New Year.

The second mechanism involves the changes which result in less dilution of the warm water core of the North Equatorial Counter Current. Once an El Niño has started, the low winds around the Equator and the collapse of the Equatorial Current mean that the diluting effects of the Ekman transport, the geostrophic return flow and tropical instability waves are all reduced in intensity.

When these two mechanisms are active, they both allow the NECC to carry warm water much further east than normal and it does so at a latitude where the atmosphere may be particularly sensitive to extra surface warming. In the cases stud-

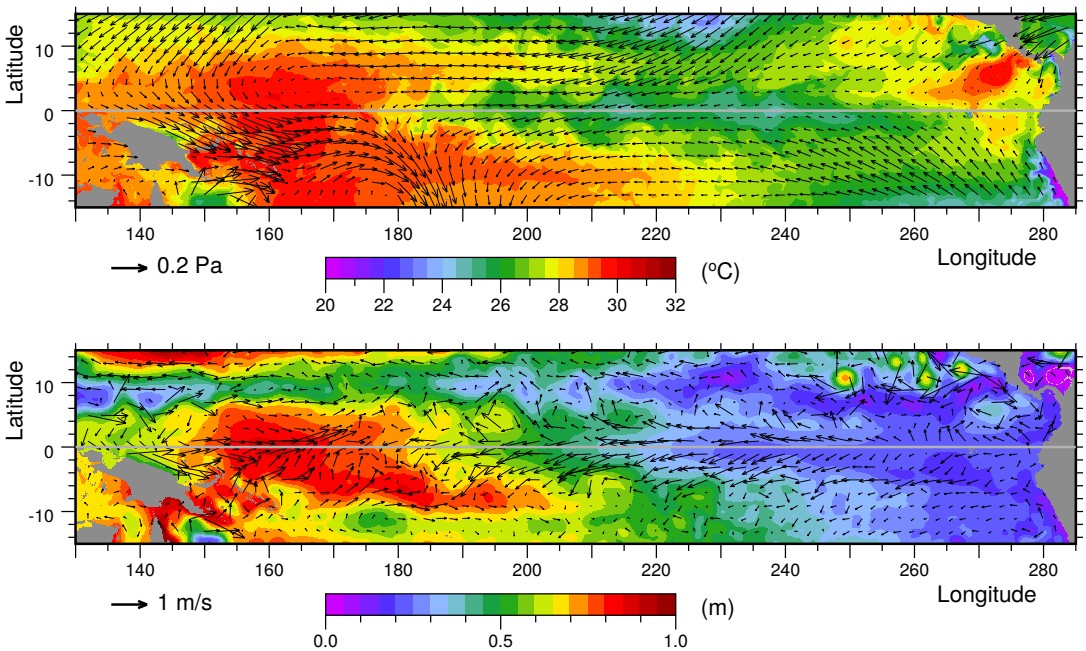

**Figure 33.** Upper: Surface temperature and wind stress vectors. Lower: Sea level (SSH) and velocity vectors from the 16th March 1997 archive dataset. Each archive dataset contains averages over the previous 5 days of the model run.

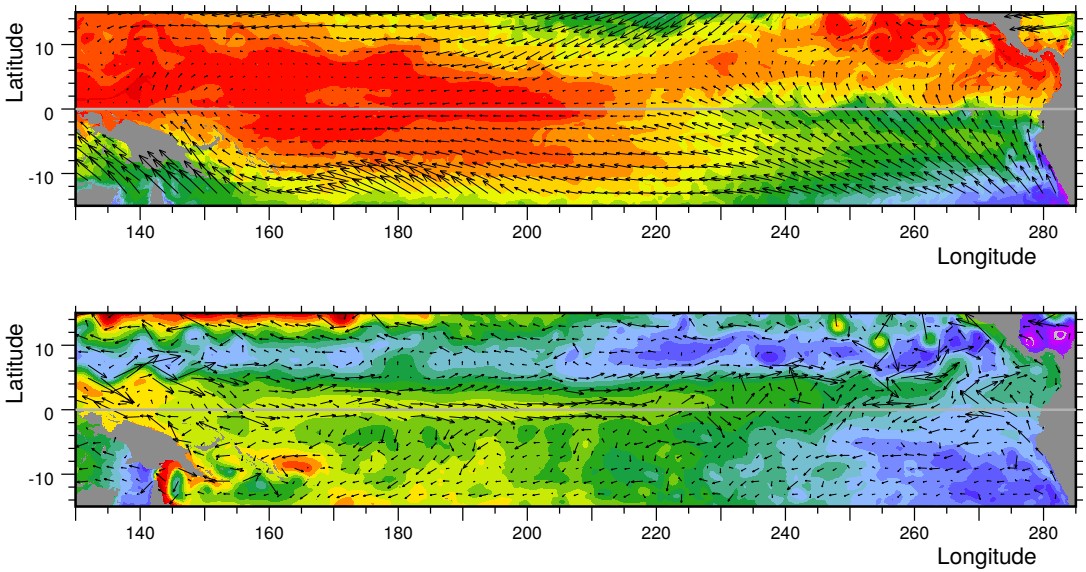

**Figure 34.** Upper: Surface temperature and wind stress vectors. Lower: SSH and velocity vectors from the 29th June 1997 archive dataset. Colours and vector scales as in Fig. 33.

ied, water with temperatures greater than 28°C was transported past the region of low winds and deep atmospheric convection to longitudes where it could trigger new episodes of deep atmospheric convection. This almost certainly had the result of extending the low wind region after which the processes can be repeated.

A third potential oceanic mechanism, that is not fully understood, involves the sea level maximum, and associated temperature maximum, that develops on the Equator in the central Pacific. In the model the maximum was generated in both strong El Niños and independently in the warm pool event of early 1997.

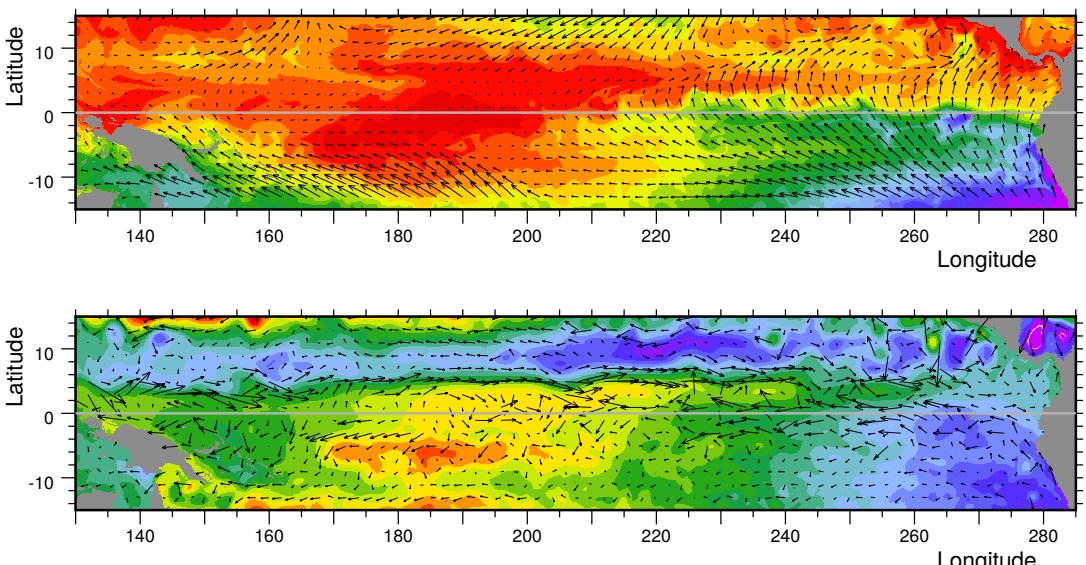

**Figure 35.** Upper: Surface temperature and wind stress vectors. Lower: SSH and velocity vectors from the 27th September 1997 archive dataset. Colours and vector scales as in Fig. 33.

It is of interest because, as discussed by Kug et al. (2009), as well as the pressure gradient along the Equator having the potential to generate currents whenever the opposing wind stress drops, the drop in sea level north and south of the Equator will result in eastward flowing geostrophic currents. These will transport warm water eastwards, independently of the local winds, and will continue as long as there is a ridge in sea level along the Equator.

Of the three mechanisms, the Rossby wave mechanism is probably of greatest importance. This is because the annual Rossby waves are generated in the eastern Pacific well before any extra advection of warm water occurs in the west. Thus a better theoretical understanding of the waves and measurements made as they first develop should allow useful predictions to be made early each year as to the probability of an El Niño.

*Code and data availability.* At the time of publication the archived data is freely available at "http://gws-access.ceda.ac.uk/public/nemo/runs/ORCA0083-N06/means/". The Nemo ocean model code and its documentation are available from "http://forge.ipsl.jussieu.fr/nemo/wiki/Users".

*Competing interests.* The author is on the advisory board of Ocean Science.

*Acknowledgements.* I wish to acknowledge the support of the Marine Systems Modelling group and the aid of Dr Andrew Coward at the UK National Oceanography Centre, part of the Natural Environment Research Council, where much of this research was carried out. I also wish to acknowledge the earlier role of the Australian CSIRO Division of Fisheries and Oceanography. Without the financial support and the professionalism and enthusiasm of staff at both centers this work would not have been possible.

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
