# Peer review of "On the Role of the North Equatorial Counter Current during a Strong El Niño"

_Ocean Science, 2017_

## Referee Comment (RC1) · Anonymous Referee #1 · 8 Feb 2018

The ms examined the role of the NECC in water mass and heat transports during El Nino events by analyzing output of a realistic model simulation. The topics is of great interest to scientific community because the origin of El Nino events are still in strong debate. Previous studies suggested roles played by warm water transfers from the western tropica Pacific to the east (McPhaden and Picaut 1990; Picaut et al. 1996; Meinen and McPhaden 2000; Jin 1997; Bunge et al. 2014). The present ms proposed the sprcific roles played by the NECC in the warm water transport frame used to explain El Nino onset. Some intersing results are obtained. However, the ms as it stands is a very preliminary draft and there are great needs to be improved. The author is asked to prepare a concise scientific paper for submission. Here are some details.

About the roles played by warm water transfers: Many excellent related papers need

to be reviewed and cited as follows. Meinen, C. S. & McPhaden, M. J. Observations of warm water volume changes in the equatorial Pacific and their relationship to El Niño and La Niña. J. Clim. 13, 3551–3559 (2000). Jin, F.-F. An equatorial ocean recharge paradigm for ENSO. Part I: conceptual model. J. Atmos. Sci. 54, 811–829 (1997) McPhaden, M. J. A 21st century shift in the relationship between ENSO SST and warm water volume anomalies. Geophys. Res. Lett. 39, L09706 (2012). Bunge, L. & Clarke, A. J. On the warm water volume and its changing relationship with ENSO. J. Phys. Oceanogr. 44, 1372–1385 (2014). McPhaden, M. J. & Picaut, J. El Niño-Southern oscillation displacements of the Western equatorial pacific warm pool. Science 250, 1385–1388 (1990). Picaut, J., Ioualalen, M., Menkes, C., Delcroix, T. & McPhaden, M. J. Mechanism of the zonal displacements of the Pacific warm pool: implications for ENSO. Science 274, 1486–1489 (1996).

A NECC-related mechanism has been proposed in the follwing papers Zhang, R.-H.* and Chuan Gao, 2016: The IOCAS intermediate coupled model (IOCAS ICM) and its real-time predictions of the 2015-16 El Niño event, Sci. Bull.ïïjŇ66 (13): 1061-1070. DOI 10.1007/s11434-016-1064-4 Zhang, R.-H. and Chuan Gao, 2016: Role of subsurface entrainment temperature (Te) in the onset of El Nino events, as revealed in an intermediate coupled model, Climate Dynamics, 46(5), 1417-1435 doi: 10.1007/s00382-015-2655-5

About the roles played by the NECC, some studies are very relevant to this study as follows Zhang, R.-H., L. M. Rothstein, A. J. Busalacchi, and X. Z. Liang, 1999: The Onset of the 1991-92 El Nino Event in the Tropical Pacific Ocean: The NECC Subsurface Pathway, Geophys. Res. Lett. , 26, 847-850. Zhang, R.-H., and A. J. Busalacchi, 1999: A possible link between off-equatorial warm anomalies propagating along the NECC path and the onset of the 1997-98 El Nino. Geophys. Res. Lett., 26,2873-2876 Zhang, R.-H., and L. M. Rothstein, 2000: The role of off-equatorial subsurface anomalies in triggering the 1991-92 El Nino as revealed by the NCEP ocean reanalysis data. J. Geophys. Res., 105, 6327-6339

The aurthor is encoueaged to read these papers carefully and then write a concise and coherent scientific paper ofr resubmission.

---

## Referee Comment (RC2) · Anonymous Referee #2 · 8 Feb 2018

General Comments The manuscript uses data from a high resolution ocean model, forced by atmospheric fields that stem from a reanalysis product, to evaluate the evolution of the 1982/83 El Nino event. The majority of the evaluation is qualitative, investigating the evolution of SSH, SST, wind stress, and current strength anomalies. Based on the analysis, hypotheses are given as to the relative role the NECC may play in the development of El Nino events. Although these hypotheses are interesting, they are not rigorously supported by the results and analysis. The paper makes bold claims about the relevance of the findings to all El Nino events, but only one El Nino event is considered through model simulations. Given that El Nino events often have distinct components to their formation, additional events and additional data (ideal some observational data) needs to be included before claims as to the applicability of these

results to all El Nino events. In addition, more quantitative analysis would be necessary to support some of the claims made, particularly about the strength of heat transport by the NECC.

Specific Comments The following areas of the paper could use the most improvement/raise the most questions

1. Consideration of a Single El Nino Event - Only the 1982/83 El Nino event is studied in this paper. While a number of other events are mentioned briefly, no rigorous analysis is performed. With only a single event considered, how can hypotheses be made that are supposed to be universal to all El Nino development? To make such claims, more El Nino events should be considered. Without these claims, I think the paper does provide an interesting analysis of a single El Nino evolution.

2. Reliance Solely on Model Data - The author acknowledges model data is imperfect, but presumably some of these ideas should be testable with model data. While data likely would not cover the 1982/83 El Nino, the TAO/TRITON buoy array should give valuable observations of the 1997/98 El Nino that could be used to verify some of the hypothesis. In addition, Argo data could provide some insight as to heat transport in the NECC during more recent El Nino events. Finally, the only proof given of the model reliability is it's ability to reproduce temperatures in the Nino regions - given that the model is used to assess more than just temperatures, a more rigorous evaluation of it's reliability would be helpful.

3. Lack of Quantification - The authors state that due to the 5 day average fields provided by the model, quantification is not possible. However, some sense of order of magnitude of quantification would be helpful, even if there are uncertainties. How much heat is on average transported by the NECC? How much does this change during the 82/83 El Nino? This would help the reader evaluate the claim as to this being a key part of the El Nino evolution. In addition, how much heat would it take to move the convection region eastward?

4. Writing Style - As is, the manuscript is challenging to read. Ideas are brought in, shelved, and returned to later on, creating a nonlinear storyline for the reader. Assertions made early on (as early as the introduction) are hard to evaluate, as none of the supporting evidence has yet been shown.

Technical Corrections Page 1, Line 17: therefore not therefor Page 2, Line 18: then not the? Page 3, Line 14: this should not be a new paragraph Page 4, Line 13: Run 6 mentioned, but no information as to what run 6 is Page 5, Line 2: 1982-83 El Nino (not 1982-82) Page 7, Line 8: a, not an Page 8, Figure 5: Boxes in figure should be labeled, as it is hard to figure out which box is which that Table 1 refers to Page 16, Line 8: the, not th Page 27, Line 8: the not th Page 33, Line 5/6: The following sentence is confusing ('as did' should be removed?): The strength of the tropical instability waves also dropped significantly as did possibly due to a reduced or reversed Equatorial Current

---

## Author Comment (AC1) · 9 Feb 2018

First I would like to thank the reviewer for reviewing the paper. This is often a thankless task but I hope that you gained something useful.

On the question of references, I really need to apologise and thank you for your list of references that I had missed. Some of them are recent but I could find only two of the earlier ones referenced, one each, in the two WOCE volumes "Ocean Circulation and Climate" edited by Siedler et al, although these contain widely quoted authoritative reviews of ocean physics and circulation.

What I think this underlines is that the role of the NECC has been neglected for too long, that it needs more work and and that it needs supporters when planning new

research initiatives such as the US TPOS 2020 program.

Your other comments really concern style. My main problem in writing this paper was that, given the importance of the region, the theory of the NECC itself, the annual Rossby waves, Tropical Instability waves and the geostrophic inflow, was surprisingly underdeveloped in the sense that it was primarily qualitative and based on model output.

For this reason I included more on things like Stommel's theory or the discussion of the low wind region than if these had been widely discussed by others. Since submitting the paper I realise I should also have at least a couple of sentences on the atmospheric profile. In the tropics these tend to lie close to the moist adiabatic of the main convection region and it is only when rising saturated air from the Central or Eastern Pacific sea surface is warmer at all levels, that deep convection can be triggered there.

I am also trying to do three things in one paper, i.e. a model analysis and two mechanisms which I have not seen proposed before (The first being the increased transport of warm water by the NECC due to the reduction in the dilution processes and the second the triggering of a stronger than normal NECC by an enhanced annual Rossby wave).

However I take your points and will try to produce a much tighter final version.

---

## Author Comment (AC2) · 9 Feb 2018

Again I would like to thank the reviewer for reviewing the paper. This is often a thankless task but I hope that you gained something useful.

In response I agree with most of the comments. The only trouble is the time required and the size of the resulting document. In the present case I thought the work had reached a stage where something should be published, enough to let people know of the results and possibly encourage (goad?) them into testing the hypotheses in other models and with observational data, including the data you mention.

I was hoping to look at later periods and/or the couple model results in a paper next year but as a compromise I will include more results in the revised paper from the

1993/94 El Nino. Neither this or any other period can prove the hypotheses but it may give confidence that they are less likely to be disproved.

For model data I really need a separate collaboration with an experimentalist. Again this can only give more confidence or disprove the hypotheses.

On the question of quantification I will have to think about what might be possible. On the question of heat transport, the transport itself is probably irrelevant as it is only water with temperatures above that required to start deep atmospheric convection (see response to Reviewer 1) that is of real interest. However I will bear the comments in mind.

Finally on writing style - I'll see what I can do!

---

## Editor Comment (EC1) · M. Hecht (Editor) · 2 Mar 2018

Dear Dr. Webb, one request that may be a firm one regards the generalization from one event to all El Ninos. That claim could be better supported with the investigation of additional El Nino events (as you propose in doi:10.5194/os-2017-99-AC2). Even then, it should perhaps be qualified as a hypothesis supported, if this proves to be the case, by the limited number of events that have been studied.

I wish you well with this and other issues that were raised, including that of style of presentation (where again there will be some expectation of accommodation).

Sincerely Yours, –Matthew Hecht

---

## Author Comment (AC3) · 12 Mar 2018

I would like to take this opportunity to thank both the reviewers and the editor for their comments and remarks which I hope will lead to a much improved paper.

I have covered most points raised in my responses to the individual reviews. However they have stimulated some further work/ideas which I plan to include in the revised paper.

1. First my use of minimum temperatures of 28C and 29C (i.e. for OSD m/s Fig. 5), was a result of my own estimate of that needed to stimulate deep convection in the atmosphere. I have since found the work of Evans & Webster (DOI:10.22499/2.6401.007) which identifies a global limit of 28C with an indication that the Western Pacific limit

may be higher.

A detailed analysis by the authors of the deep convection regions affecting the El Nino would be useful.

2. Following the suggestion of referee 2, I have investigated the model behaviour during the 1997-98 El Nino in more detail. The results indicate that the NECC is again transporting warm water into the eastern Pacific during the autumn of 1997 for the same reasons it does in 1982.

However as is indicated in (OSD m/s) Fig 4, the central Pacific is also warmed during the spring of 1997, something that did not happen in 1982. The forcing datasets indicate that the average strength of what might be called the New Guinea (or Bismark Sea) Atmospheric Jet (viz. the East African Jet) was stronger in the period 1996.9 to 1997.4 than in the same period during the previous two years - so this may be responsible for the systematic changes seen in early 1997.

I will have to deal with these points in the revised paper but the whole thing really needs a separate publication.

Regards,

David Webb

---

## Author Response (AR1)

Author's response

OS-2017-99
On the Role of the North Equatorial Counter Current during a Strong El Niño
David J. Webb

1.  Analysis of the 1997-1998 El Niño

Following the comments of reviewer 2, I have carried out a similar analysis of the strong 1997-1998 El Niño.  This appears to start differently but the main results of the paper are confirmed, the NECC having an important role in transporting warm water, especially in the eastern Pacific.  There was also a similar reduction in the role of Ekman transport, the geostrophic inflow and tropical instability waves.  There was also a stronger than normal annual Rossby wave which increased the speed and transport of the NECC.

The additional analysis resulted in a significant increase in the size of the paper.  I have tried to keep this to a minimum by combining figures and keeping the text short.

2.  Style

Following the comments of Reviewer 1, I have changed the style of presentation especially in the introduction and discussion sections.  All mention of hypotheses has gone and as far as possible it is a straight description of what was really a sequential series of numerical experiments and their results.

One remaining weakness of the paper is the necessity of guessing what the atmosphere is doing at each stage during the development of an El Niño.  I have discussed this with a senior UK meteorologists who has worked on the El Nino problem and apparently this overview data is not available - which is why the Climate community still makes widespread use of SST values.

3.  Overall

I have changed the title of the paper slightly and typographical errors in the discussion paper.  In addition to the changes referred to above I have also made many small changes throughout the paper to make it easier to read.

Regards,

David Webb
23 April 2018

[revised manuscript text omitted]

*Copyright statement.* The works published in this journal are distributed under the Creative Commons Attribution 4.0 License. This licence does not affect the Crown copyright work, which is reusable under the Open Government Licence (OGL). The Creative Commons Attribution 4.0 License and the OGL are interoperable and do not conflict with, reduce or limit each other.

*Acknowledgements.* I wish to acknowledge the support of the Marine Systems Modelling group and the aid of Dr Andrew Coward at the UK National Oceanography Centre, part of the Natural Environment Research Council, where much of this research was carried out. I also wish to acknowledge the earlier role of the Australian CSIRO Division of Fisheries and Oceanography. Without the financial support and the professionalism and enthusiasm of staff at both centers this work would not have been possible.

---

## Referee Report (RR1)

The paper has greatly improved with this revision – specifically the writing has made the flow of ideas much easier to follow and the addition of the 1997/98 El Nino event has supported the argument that the physics were not restricted to just one event. There are still a few questions I'd like to see addressed, specifically related to model validation and ignoring the impact of radiation on equatorial temperatures.

The author validates the model with SST only, but there are many other components of the physics important for claims. Reproducing SST is only an accurate assessment of these other processes if the model captures all the physics correctly. Particularly important for the paper's claims is the model's ability to capture subsurface transport. Some discussion as to the model's ability to accurately represent the currents and their variability (or why that can't be evaluated) would be helpful.

The author claims that there isn't enough heat locally to explain warming signal – but how is that determined? It seems as though the only heat flux considered to explain warming is from ocean advection, but it is unclear why radiation cannot add heat to the equatorial Pacific. Why was this neglected when considering heat fluxes? Also, why couldn't a reduction in cold-water flux also contribute to warming? Some justification as to why only an increase in warm water flux considered would help understanding.

There is a brief discussion as to how EUC flow rates would reduce tropical instability wave generation (TIW). The authors could be more quantitative here, using the Richardson number to get a sense of how stress changes impact the instability (see Moum et al, 2009).

Minor Comments:

Page 2, Line 31: missing 'is'. Should read: "…model, it is likely that this **is** one, or possibly the main, factor…"
Page 3, Line 36: the idea of 'run 6' is introduced, but it's confusing as to why run 6 is significant. How many runs are there? What is the difference with the other runs? Is that just a name of one of the pre-run Nemo simulations?
Page 4, Line 30: atmosphere is misspelled
Page 12, Line 14: 'the' misspelled
Page 14, Line 11: typo? "…is reduced in the ocean to north **and west the west**."
Page 14, Line 14-16: Grammar issues. Not sure if this is supposed to be one sentence of two with capitalization/verb issues: "…as it continues eastwards it has the potential to trigger deep atmospheric convection. further convection, thus moving the region of atmospheric convection steadily eastwards. "
Page 17, Line 31: Missing N: "By the end of June the annual wave at **6°**has…"
Page 26, Line 45: Repeated word: "in less dilution of the warm water **water** core of the North"

---

## Author Response (AR2)

**Response to Comments**

I would like to thank the reviewers for taking time with this manuscript and for their helpful comments.

**Reviewer's comments in bold** *My response in italic.* Changes to the paper in normal text.**

**Reviewer 2**

The paper has greatly improved with this revision – specifically the writing has made the flow of ideas much easier to follow and the addition of the 1997/98 El Nino event has supported the argument that the physics were not restricted to just one event. There are still a few questions I'd like to see addressed, specifically related to model validation and ignoring the impact of radiation on equatorial temperatures.

The author validates the model with SST only, but there are many other components of the physics important for claims. Reproducing SST is only an accurate assessment of these other processes if the model captures all the physics correctly. Particularly important for the paper's claims is the model's ability to capture subsurface transport. Some discussion as to the model's ability to accurately represent the currents and their variability (or why that can't be evaluated) would be helpful.

The main reasons that validation stopped at this stage are:

(a) The underlying model code and parameterisations had already been validated many times through the use of the model in oceanic and climate research.

(b) The extensive and validated observational SST dataset was in itself a very good test of the model.

(c) At the time, when I had no idea which other aspects of the ocean circulation might prove critical,

the cost of extra validation studies did not appear to justify the limited additional confidence gained. (i.e. I did not want to waste my own time and effort – which is all that is available).

Given the results of the paper there is now a case for further studies of Tropical Instability Waves and the Annual Rossby Waves including comparisons between the model and observations.

On this basis I have changed the final part of the relevant paragraph:

... closest to the ocean surface.

On the basis of this analysis and the additional confidence in the model code which comes from its successful use over many years for oceanic and climate research, it appears reasonable to make use of the model archive data in the present study of the processes affecting the El Niño. OK

I have also removed some repetitive text from the start of the previous paragraph and in the following paragraph I have added a section on the diurnal variation.

The author claims that there isn't enough heat locally to explain warming signal – but how is that determined? It seems as though the only heat flux considered to explain warming is from ocean advection, but it is unclear why radiation cannot add heat to the equatorial Pacific. Why was this neglected when considering heat fluxes? Also, why couldn't a reduction in cold--

**water flux also contribute to warming? Some justification as to why only an increase in warm water flux considered would help understanding.**

The validation paper does discuss the surface flux fields in Nino regions 1(Fig. 11) and 3.4 (Fig. 13), the main ones of interest to the 'classical oceanic El Nino researchers' and the 'modern meteorological community'. The calculations show that during the second half of 1982 when the El Nino was developing the net flux into Nino region 1 was less than during the previous year. In Nino region 3.4 the flux into the ocean was much less than in the previous year. For short wavelength radiation by itself, the fluxes were about the same in Nino region 1 but less during the development of the El Nino in Nino region 1.

On this basis the surface fluxes cannot explain the increases in SST along the Equator. It is possible that the model physics is completely wrong and that reduced subsurface mixing or reduced upwelling explains the observed increases in SST but this is unlikely.

So paragraph added at end of section 3 immediately before section 4.

An increase in the local surface heat flux might produce the observed increases in model SST. However Webb (2016, see Figs. 11 and 13) investigated these fluxes in two of the key Nino regions and found that in each case the net heating was less during the development of the 1982-1983 El Nino than in the two previous years. Significant errors in the models vertical mixing scheme could also be responsible but this is unlikely. Instead it is much more probable that the increases in both the model SST and observations are due to advection of heat by the ocean.

**There is a brief discussion as to how EUC flow rates would reduce tropical instability wave generation (TIW). The authors could be more quantitative here, using the Richardson number to get a sense of how stress changes impact the instability (see Moum et al, 2009).**

I do not really understand this. As discussed in the text TIW appear to be due to barotropic or baroclinic instabilities. Richardson number is more associated with vertical shears. Following the paper of Moum et al. (2009) it is possible that the reviewer meant that the increased vertical shear due to TIWs also cools the NECC during non-El Nino years. However if I was to investigated the suggestion properly at this time I would require a repeated run for which there are no resources.

This is because the model only archives average temperature and average vertical mixing coefficient. During individual model timesteps the vertical mixing coefficient can be orders of magnitude greater than normal when the vertical temperature gradient approaches zero. I found that such events appear to dominate the calculation of the average mixing coefficient but if this is then used with the vertical gradient of the averaged temperature, to calculate a vertical flux, the values are unrealistically large.

So I am going to ignore this one. I understand that if the paper is accepted, then this discussion will be published, so the reviewer can be credited with this additional suggested mechanism.

However the Moum et al 2009 paper gives observational support to the model's analysis of TIW strength during the 1995-2000 period. I have therefore added immediately before section 9.2

The plot by Moum et al. (2009, fig 4) based on mooring data from the Equator at 220E(140W), shows a similar reduction in TIW strength during 1997 and 1998.

**Minor Comments:**

Page 2, Line 31: missing 'is'. Should read: "...model, it is likely that this is one, or possibly the main, factor..." Corrected

Page 3, Line 36: the idea of 'run 6' is introduced, but it's confusing as to why run 6 is significant. How many runs are there? What is the difference with the other runs? Is that just a name of one of the pre--run Nemo simulations?

A new paragraph has been inserted and the start of the paragraph changed to read:

This high resolution version of the Nemo ocean model was developed for coupling with a similar high-resolution version of the UK Meteorological Office atmospheric model, the aim being to create an improved coupled model for both weather prediction and climate change research. However before coupling the two models, a number of test runs were carried out using just the ocean model. The data analysed here comes from run 6, the last and longest of these tests.

The surface boundary conditions used for run 6 are those of ...

**Page 4, Line 30: atmosphere is misspelled. Corrected.**

Page 12, Line 14: 'the' misspelled. Corrected.

Page 14, Line 11: typo? "...is reduced in the ocean to north and west the west." Page 14, Line 14--16: Grammar issues. Not sure if this is supposed to be one sentence of two with capitalization/verb issues: "...as it continues eastwards it has the potential to trigger deep atmospheric convection. further convection, thus moving the region of atmospheric convection steadily eastwards. "

The paragraph concerned has been changed to:

However in an El Ni\~{n}o year, once the region of low wind stress has started moving eastwards, the strength of these processes is reduced in the ocean to its north and west. As a result the core of the NECC passing the eastern boundary of the low wind region is much warmer than normal and as it continues eastwards it has the potential to trigger new episodes of deep atmospheric convection. As a result the region of deep atmospheric convection may progress steadily eastwards.

[revised manuscript text omitted]

Depth